# NPM1 upregulates the transcription of PD-L1 and suppresses T cell activity in triple-negative breast cancer

Ge Qin[1,2,5], Xin Wang[1,5], Shubiao Ye[2,5], Yizhuo Li[1,5], Miao Chen[1,5], Shusen Wang[1], Tao Qin[3], Changlin Zhang[1], Yixin Li[1], Qian Long[1], Huabin Hu[2], Dingbo Shi[1], Jiaping Li[1], Kai Zhang[1], Qinglian Zhai[1], Yanlai Tang [4], Tiebang Kang [1], Ping Lan [2], Fangyun Xie[1], Jianjun Lu[4] & Wuguo Deng[1✉]

Programmed cell death protein-1 (PD-1)/programmed cell death ligand-1 (PD-L1) interaction plays a crucial role in tumor-associated immune escape. Here, we verify that triple-negative breast cancer (TNBC) has higher PD-L1 expression than other subtypes. We then discover that nucleophosmin (NPM1) binds to *PD-L1* promoter specifically in TNBC cells and activates *PD-L1* transcription, thus inhibiting T cell activity in vitro and in vivo. Furthermore, we demonstrate that PARP1 suppresses *PD-L1* transcription through its interaction with the nucleic acid binding domain of NPM1, which is required for the binding of NPM1 at *PD-L1* promoter. Consistently, the PARP1 inhibitor olaparib elevates *PD-L1* expression in TNBC and exerts a better effect with anti-PD-L1 therapy. Together, our research has revealed NPM1 as a transcription regulator of *PD-L1* in TNBC, which could lead to potential therapeutic strategies to enhance the efficacy of cancer immunotherapy.

[1] Sun Yat-sen University Cancer Center, State Key Laboratory of Oncology in South China, Collaborative Innovation Center of Cancer Medicine, Guangzhou, China. [2] The Sixth Affiliated Hospital, Sun Yat-sen University, Guangzhou, China. [3] Sun Yat-sen Memorial Hospital, Sun Yat-sen University, Guangzhou, China. [4] The First Affiliated Hospital, Sun Yat-sen University, Guangzhou, China. [5] These authors contributed equally: Ge Qin, Xin Wang, Shubiao Ye, Yizhuo Li, Miao Chen. ✉email: dengwg@sysucc.org.cn

Triple-negative breast cancer (TNBC) constitutes about 10–20% of all breast cancer[1]. TNBCs are generally associated with advanced stage, higher tendency to metastasize and comparatively short survival[2]. In the past decade, advances in endocrine therapy and anti-human epidermal growth factor receptor 2 (HER2) therapy have remarkably improved the survival of estrogen receptor (ER) positive and HER2 positive breast cancer patients. However, effective therapeutic strategies for TNBC are still desperately in need.

Immunotherapy has been proved to be a promising treatment for TNBC in recent years. Studies have shown that tumor-infiltrating lymphocytes (TILs) are associated with improved prognosis in TNBC patients[3,4], and TNBC is the most immunogenic subtype. TNBCs have higher expression of PD-L1, which is an immune checkpoint molecule and an important target in immunotherapy[5,6]. The PD-1/PD-L1 axis generates an inhibitory signal that attenuates the activity of T cells and contributes to tumor immune escape[7,8]. In breast cancers, PD-L1 high/positive expression is associated with larger tumor size, higher tumor grade, increased positive lymph node number, as well as negative ER and progesterone receptor (PR) status[9,10]. According to the outcomes in clinical trials, the response rate of anti-PD-L1 monotherapy ranges from 5 to 20%, and the percentage is higher in patients with positive PD-L1 expression[11].

The expression of PD-L1 can be exogenously induced by various cytokines including IFN-γ, EGF, interleukins (ILs) and TNF-α, and the JAK/STAT1/IRF1, PI3K/AKT/mTOR, NF-κB and JAK/STAT3 signaling pathways regulate the activation and nuclear translocation of the downstream transcription regulators to enhance PD-L1 transcription[12,13]. Besides, intrinsic carcinogenic changes can induce PD-L1 expression. For instance, transcription factor AP-1 promotes the expression of PD-L1 in Hodgkin lymphomas by binding to the AP-1-responsive enhancer in the PD-L1 gene[14], and HIF-2α targeted PD-L1 in renal cell carcinoma[15]. In TNBC, the protein expression and mRNA level of PD-L1 are higher than other subtypes. It has been reported that PTEN loss increased PD-L1 transcription in TNBC cells[16], while CMTM6 promoted PD-L1 protein half-life and cell surface expression[17]. Moreover, glycogen synthase kinase 3β (GSK-3β) has been demonstrated to interact with PD-L1 to induce its degradation[18]. Nevertheless, the exact transcriptional regulation of PD-L1 in TNBC remains largely controversial.

Nucleophosmin (also known as NPM1 or B23) is a highly abundant protein crucial for multiple cellular functions, including ribosome biogenesis, chromatin remodeling, centrosome duplication, embryogenesis, apoptosis and DNA repair[19]. The structural architecture of NPM1 is mainly characterized into three distinct regions: the well-conserved N-terminal domain that mediates NPM1 oligomerization and interactions with other proteins, the acidic domains in the center for histone binding, and the C-terminal nucleic acid binding domain[20].

The oncogenic role of NPM1 is mainly reported in acute myeloblastic leukemia (AML). Thirty-five percent of all AML patients are diagnosed with NPM1 rearrangements or mutations[21]. Though there is little evidence of NPM1 mutation in solid tumors[22], the wild type NPM1 is overexpressed in various tumors. NPM1 promotes metastasis in colon cancer and serves as a poor prognostic factor[23]. High expression of NPM1 is associated with tumorigenesis in bladder urothelial carcinoma[24]. Besides, downregulation of NPM1 increases radiation sensitivity in non-small-cell lung cancer (NSCLC)[25]. In addition, NPM1 has been shown to facilitate the DNA binding activity of NF-κB and upregulates the NF-κB-mediated transcription[26]. Nonetheless, the immune regulation activity of NPM1 in cancer has not been reported.

In this study, we verify that PD-L1 is highly expressed on both mRNA and protein levels specifically in TNBCs, and identify NPM1 as a transcription activator of PD-L1. We further demonstrate that PARP1 suppresses PD-L1 expression via interaction with NPM1, which abolish its binding at PD-L1 promoter in TNBCs. Supporting this regulation mechanism, our experiment in orthotopic breast cancer mouse model shows that PD-L1 and PARP inhibitor combination therapy has better effects than monotherapy in the treatment of TNBC. Collectively, our study has revealed the regulatory role of NPM1 in immune escape mediated by PD-L1 in TNBC, which suggests that NPM1 is a potential target for TNBC treatment.

## Results

**TNBCs have higher PD-L1 expression**. PD-L1 protein expression was examined in 149 breast cancer patients by immunohistochemical staining (Fig. 1a). Pearson chi-square analysis was used to determine the correlation between PD-L1 expression and other clinical features. PD-L1 positive rate in TNBC was 61.5% (32/52), but was only 18.6% (18/97) in non-TNBC (Fig. 1b and Supplementary Table 1). In addition, tumors in larger volume (diameter > 20 mm) had a higher positive rate, which was in significant inverse correlation with hormone receptor (HR) status (Supplementary Table 1). Survival analysis showed that the overall survival (OS) of PD-L1 positive patients and PD-L1 negative patients had no significant difference in the whole cohort (Fig. 1c, left; Supplementary Table 2). However, PD-L1 positive patients had remarkably shorter OS in subgroup analysis for TNBC (Fig. 1c, right panel). We also analyzed the Kaplan Meier survival for PD-L1 in early stage (phase I) and middle stage (phase II–III) breast cancer patients. The result showed that PD-L1 was associated with shorter OS in early stage patients, but such a correlation was not observed in middle stage patients. (Supplementary Fig. 1A). Consistently, PD-L1 mRNA level was higher in TNBC according to TCGA database (Fig. 1d). Moreover, in a panel of breast cancer cell lines that contained five TNBC cell lines (MDA-MB-231, HCC1937, BT20, HCC1806, and HS578T) and three non-TNBC cell lines (MCF-7, T47D and SKBR3), PD-L1 was found to have higher protein and mRNA levels in TNBC cell lines (Fig. 1e, f). According to these results, we inferred that specific regulation mechanism of PD-L1 transcription might exist in TNBC.

**NPM1 binds to PD-L1 promoter and increases PD-L1 expression**. To identify potential transcription regulators of PD-L1 in TNBC, we constructed luciferase reporter plasmids bearing different lengths of the core promoter region of PD-L1 (Fig. 2a). Dual luciferase reporter assay suggested that the −690 ~ −24 bp region was most critical to the activity of PD-L1 promoter (Fig. 2b). Using a biotin-labeled DNA probe with this region of PD-L1 promoter, we pulled down nuclear protein-DNA complexes by streptavidin-agarose beads in both TNBC and non-TNBC cells (Fig. 2c). Proteins associated with the PD-L1 promoter probe were separated by SDS-PAGE and silver-stained. As shown in Fig. 2d, a protein band around 35 kDa was enriched in TNBC cells but not in non-TNBC cells. Subsequent analysis of the band by mass spectrometry revealed that NPM1 was a candidate PD-L1 promoter binding protein (Supplementary Fig. 1B). To validate this finding, chromatin precipitation (ChIP) was performed, and NPM1 was found to mainly bind at the −690 ~ −469 bp region of PD-L1 promoter (Fig. 2e).

We initially used five TNBC cell lines to analyze the basic expression of NPM1. Among them, HS578T is a fibroblast-like cell line, HCC1937 is a BRCA1 mutant cell line, and BT20 typically over-expresses WNT3 and WNT7B oncogenes. To avoid

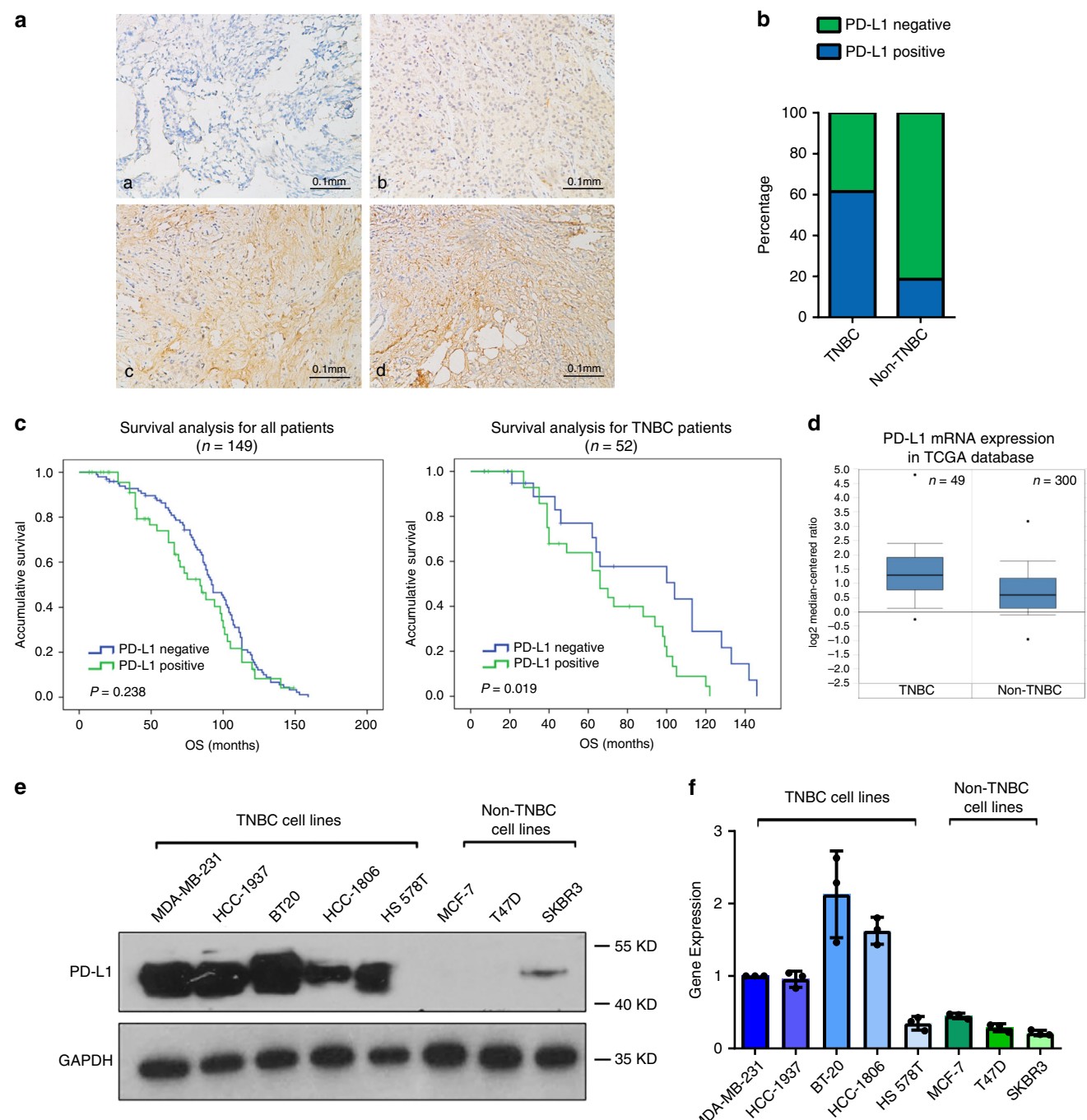

**Fig. 1 TNBCs have higher PD-L1 expression. a** Representative images of immunohistochemical (IHC) staining of PD-L1 in 149 breast cancer tissues (**a**, PD-L1 negative; **b**–**d**, PD-L1 positive). **b** PD-L1 positive rate in 52 TNBC and 97 non-TNBC patient samples. Tow-side Pearson Chi-square analysis was used to determine the correlation. **c** Kaplan–Meier analysis for all breast cancer subtypes and TNBC based on *PD-L1* expression. Data was analyzed by log-rank test. **d** *PD-L1* mRNA expression in TNBC and non-TNBC in TCGA database ($P < 0.0001$). For TNBC group, maximum = 4.806, median = 1.28, minimum = −0.265, upper bound (75th percentile) = 1.913, lower bound (25th percentile) = 0.766. For non-TNBC group, maximum = 3.176, median = 0.603, minimum = −0.973, upper bound (75th percentile) = 1.173, lower bound (25th percentile) = 0.129. **e** Western blot analysis of PD-L1 in TNBC cell lines (MDA-MB-231, HCC1937, BT20, HCC1806, and HS578T) and non-TNBC cell lines (MCF-7, T47D, and SKBR3). **f** RT-qPCR analysis of *PD-L1* mRNA in TNBC and non-TNBC cell lines. Data were presented as mean ± s.d. of three independent experiments. Source data are provided as a Source Data file.

potential effects caused by strong genetic heterogeneity, the MDA-MB-231 and HCC1806 cell lines were used for further verification. Functionally, *PD-L1* promoter activity was decreased when NPM1 was knocked down by small interfering RNA (si-RNA) in MDA-MB-231 cells, but enhanced by NPM1 over-expression (Fig. 2f). Irrelevant si-RNA (si-APCL) and promoter (FOP promoter) were used as controls to confirm the regulatory

activity of NPM1 was specific (Supplementary Fig. 1C). Besides, both the mRNA and protein levels of PD-L1 were downregulated in MDA-MB-231 and HCC1806 cells with NPM1 knocked down, while NPM1 overexpression upregulated PD-L1 expression (Fig. 2g, h). Considering that interferon gamma (IFN-γ) had been reported to induce PD-L1 expression in many tumor cells, we investigated whether it was also the case in stable TNBC cell

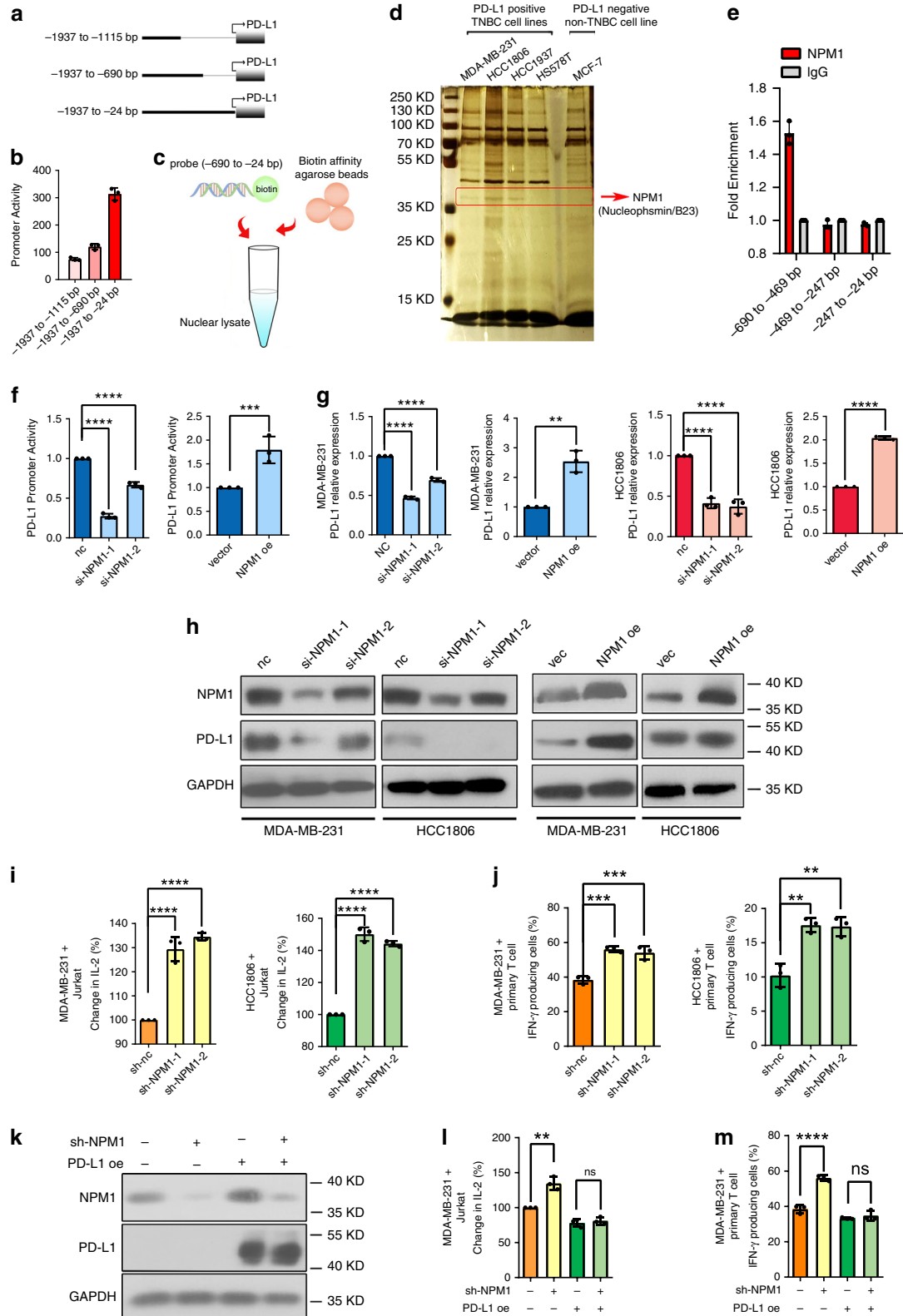

lines with *NPM1* silenced by lentiviral short-hairpin RNA (shRNA) and downregulated PD-L1. As shown in Supplementary Fig. 1D, knockdown of NPM1 also decreased PD-L1 expression in the presence of IFN-γ, suggesting that NPM1 was a dominant regulatory factor for PD-L1 expression in TNBC cells.

Flow cytometry demonstrated that knockdown of NPM1 caused decreased expression of PD-L1 on the surface of TNBC

cells (Supplementary Fig. 1E). To further investigate the function of NPM1 in immune regulation, MDA-MB-231 and HCC1806 TNBC cells were co-cultured with Jurkat cells[17,18,27,28]. ELISA results showed that knockdown of NPM1 in TNBC cells promoted the secretion of interleukin-2 (IL-2) by Jurkat cells (Fig. 2i). Consistently, *NPM1* silencing in MDA-MB-231 and HCC1806 cells increased the percentage of primary T cells that

**Fig. 2 NPM1 binds to *PD-L1* promoter and upregulates PD-L1 expression in TNBC cells. a** Fragments of *PD-L1* promoter in the luciferase reporter plasmids. **b** Activity of the different fragments of *PD-L1* promoter was determined by dual-luciferase assay. $n = 3$ independent experiments. **c** Schematic diagram of the streptavidin-agarose pull-down assay. **d** Proteins from the pull-down assay were separated by polyacrylamide gel eletrophoresis and revealed by sliver staining. The protein band marked in red rectangle was cut out to be analyzed by mass spectrometry. Three biological replicate samples were used. **e** Binding of NPM1 to *PD-L1* promoter was examined by ChIP-qPCR. $n = 3$ independent experiments. **f** *PD-L1* promoter activity in MDA-MB-231 cells transiently transfected with control siRNA (nc), *NPM1* siRNAs (si-NPM1-1, si-NPM1-2), control plasmid for NPM1(vector), or NPM1 overexpression plasmid (NPM1 oe) was measured by dual-luciferase assay. **g, h** *PD-L1* mRNA levels (**g**), NPM1 and PD-L1 protein expression (**h**) in MDA-MB-231 and HCC1806 cells with transient NPM1 knockdown or overexpression was detected. **i** Jurkat cells were co-cultured with MDA-MB-231 or HCC1806 cells in a 1:4 ratio for 48 h, and IL-2 secreted was detected by ELISA. **j** Primary T cells were co-cultured with MDA-MB-231 and HCC-1806 cells in a 1:30 ratio for 24 h. IFN-γ producing T cells were detected by flow cytometry. **k** Western blot analysis of NPM1 and PD-L1 expression in MDA-MB-231 cells with NPM1 knockdown and/or PD-L1 overexpression. **l, m** MDA-MB-231 cells with NPM1 knockdown and/or PD-L1 overexpression were co-cultured with Jurkat cells (**l**) or primary T cells (**m**). Data were presented as mean ± s.d. of three independent experiments. **f** Right panel, **g** 2th and 4th panel were analyzed by two-side Student's *t*-test, \*\**P* < 0.01, \*\*\**P* < 0.001, \*\*\*\**P* < 0.0001. **f** Left panel, **g** 1th and 3th panel, **i, j** were analyzed by one-way ANOVA + two-side Dunnett test, \*\**P* < 0.01, \*\*\**P* < 0.001, \*\*\*\**P* < 0.0001. **l, m** were analyzed by one-way ANOVA + two-side Tukey test, \*\**P* < 0.01, \*\*\*\**P* < 0.0001; ns not significantly different. Source data are provided as a Source Data file.

produced IFN-γ (Fig. 2j). Using murine B16-OVA cells, we also demonstrated that knockdown of NPM1 suppressed the expression of PD-L1 (Supplementary Fig. 1F), and stimulated IFN-γ production by the co-cultured splenic lymphocytes separated from OT-I mouse (Supplementary Fig. 1G). Moreover, PD-L1 was overexpressed in MDA-MB-231 cells with NPM1 knocked down (Fig. 2k), and the elevated secretion of IL-2 by co-cultured Jurkat cells as well as the production of IFN-γ in co-cultured primary T cells were fully reversed (Fig. 2l, m). Altogether, our experiments indicated that high expression of NPM1 in TNBC cells inhibited T lymphocyte activity through up-regulating PD-L1 expression.

**Knockdown of NPM1 suppresses PD-L1 expression in vivo.** To confirm the function of NPM1 in immunosuppression in vivo, stable mouse 4T1 breast cancer cells without or with NPM1 knockdown or/and PD-L1 overexpression were constructed (Supplementary Fig. 2D). $0.5 \times 10^6$ 4T1 cells were orthotopically injected into the mammary fat pads of female BALB/c immunocompetent mice, and the tumor volume in each mouse was measured twice a week. Our results showed that NPM1 knockdown significantly decreased the tumor size (Fig. 3a and Supplementary Fig. 2E), and reduced the occurrence rate of pulmonary metastasis as well as the number of nodules formed in lungs (Fig. 3b; Supplementary Fig. 3A, B). It was noteworthy that knockdown of NPM1 had no effect on the viability and clone formation ability of 4T1 cells in vitro in the absence of lymphocytes (Supplementary Fig. 2A–C), which implied that the smaller size of tumors formed by NPM1 knockdown cells was probably due to the enhanced activity of immune cells in the microenvironment. The interaction of PD-1 expressed on CD8+ T cells and PD-L1 expressed on tumor cells plays a key role in immune escape. The activation of the PD-1/PD-L1 axis restricts CD8+ T cell expansion and inhibits its anti-tumor activity. In line with this deduction, tumor-infiltrating lymphocytes were analyzed by flow cytometry, and the CD45+ CD8+ T cell population was remarkably increased in tumors with *NPM1* silenced (Fig. 3c, top and Supplementary Fig. 3C). Also, the percentage of active CD8+ T cells expressing CD107 and CD69 were significantly increased in tumors with NPM1 downregulated (Fig. 3c, middle and bottom and Supplementary Fig. 3D, E). The gating strategy for CD45+ cells was shown in Supplementary Fig. 3F. In addition, the increased ratio and activity of CD8+ T cells caused by NPM1 knockdown were reversed by PD-L1 overexpression. Immunofluorescence (IF) microscopy showed that NPM1 knockdown promoted CD8+ lymphocyte infiltration and granzyme B secretion in tumor tissues (Fig. 3d). IHC staining of the tumor tissues demonstrated that the expression of PD-L1 was

decreased when NPM1 was knocked down (Fig. 3e), supporting that NPM1 modulated T cell activity by regulating PD-L1 expression in mouse model.

**NPM1 is a poor prognostic factor in breast cancer.** To further confirm the prognostic value of NPM1 in breast cancer, NPM1 expression was detected by IHC in a tissue microarray containing 133 breast cancer patient samples (Fig. 4a). The Kaplan–Meier survival curves showed that breast cancer patients with NPM1 high expression had shorter OS, which indicated that NPM1 was a poor prognostic factor in breast cancer (Fig. 4b). Consistently, analysis of NPM1 mRNA in Kaplan–Meier Plotter database also suggested that NPM1 was a poor prognostic factor especially in TNBC (Supplementary Fig. 4B). Although NPM1 was not an independent prognostic factor based on our COX-regression analysis (Supplementary Table 3), our analysis showed that patients with negative hormone receptor expression tended to have higher level of NPM1 (Supplementary Table 4), suggesting that NPM1 still had prognostic value in breast cancer treatment.

Moreover, a higher percentage of TNBC patients had elevated NPM1 expression (Fig. 4c), which was in agreement with our result that NPM1 expression was higher in most TNBC cell lines (Supplementary Fig. 4A). Additionally, NPM1 expression was positively correlated with PD-L1 expression in breast cancer patients (Fig. 4d).

**PARP1 interacts with NPM1 and inhibits *PD-L1* transcription.** In order to further delineate the molecular mechanism of NPM1-regulated *PD-L1* transcription, NPM1 was immunoprecipitated in MDA-MB-231 cells, and its binding proteins were analyzed by mass spectrometry (Fig. 5a). Intriguingly, PARP1 was among candidate proteins that interacted with NPM1 (Supplementary Fig. 5A), and was predicted to be a transcription regulator of PD-L1 by GCBI database (Supplementary Fig. 5B). To investigate the functional link between NPM1 and PARP1, we first validated the interaction between Flag-tagged NPM1 and Myc-tagged PARP1 in HEK293T cells by reciprocal co-IP (Fig. 5b). Next, the interaction between endogenous NPM1 and PARP1 was confirmed by reciprocal co-IP in MDA-MB-231 cells (Fig. 5c). Moreover, Flag-NPM1 and Myc-PARP1 were purified and treated with RNase for IP in a cell-free system, which verified that the two proteins interacted directly (Fig. 5d and Supplementary Fig. 5C). Finally, immunofluorescence (IF) was performed to show that NPM1 and PARP1 were co-localized in cell nuclei (Supplementary Fig. 5D).

To examine the role of PARP1 in PD-L1 transcriptional regulation, luciferase reporter assay showed that PARP1 knockdown increased *PD-L1* promoter activity, whereas PARP1 overexpression suppressed *PD-L1* promoter activity (Fig. 5e).

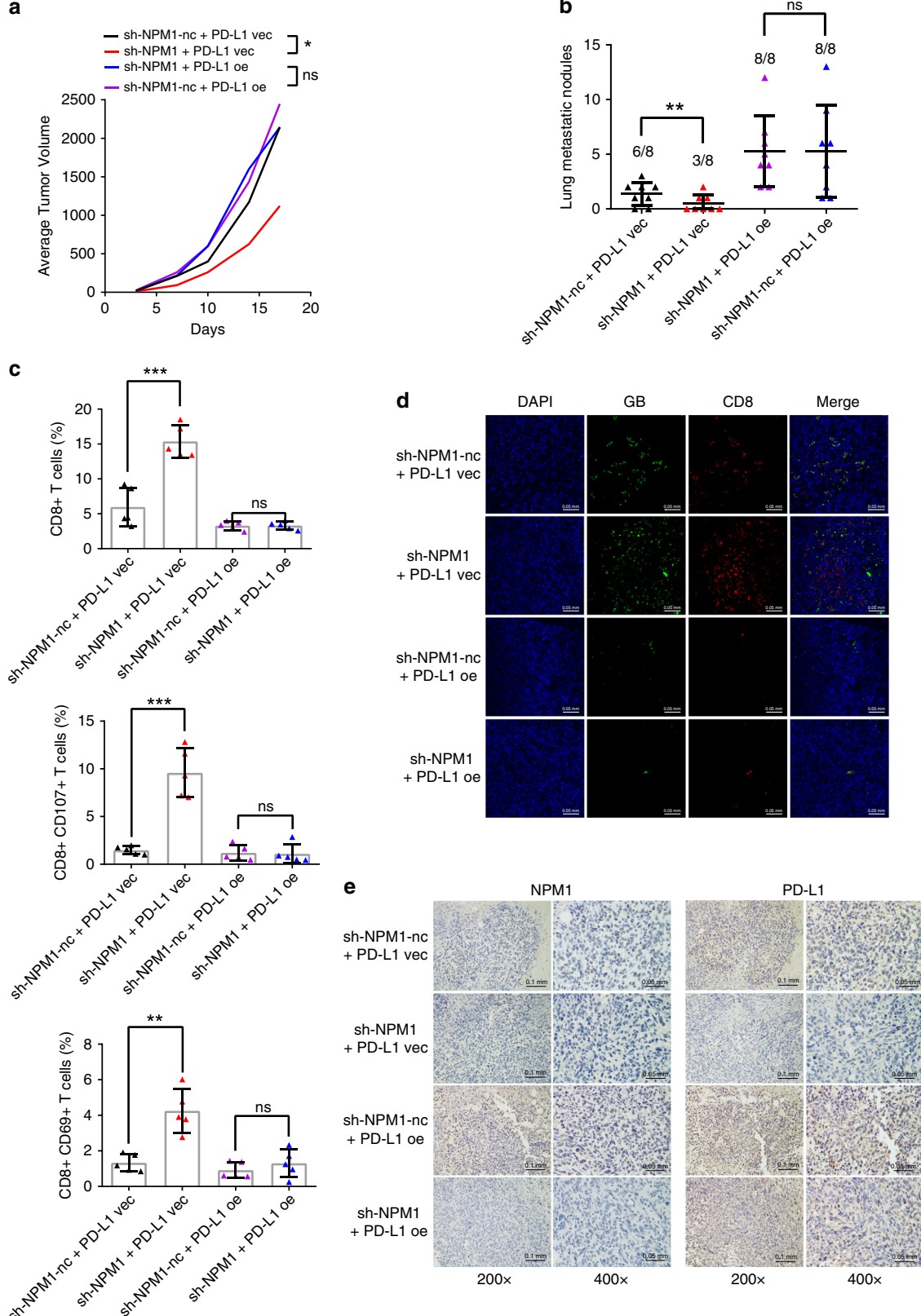

Furthermore, knockdown of PARP1 increased the mRNA and protein levels of PD-L1 in both MDA-MB-231 and HCC1806 cells, while PARP1 overexpression had the opposite effects (Fig. 5f, g). Likewise, stable cell lines with *PARP1* silenced had a larger amount of cellular PD-L1 (Supplementary Fig. 5E) as well as increased expression of PD-L1 on cell surface (Supplementary Fig. 5F), and

dramatically repressed the secretion of IL-2 by co-cultured Jurkat cells (Fig. 5h). Consistently, the PARP inhibitor olaparib was shown to enhance the promoter activity of *PD-L1*, and upregulate its expression (Fig. 5i). Together, these in vitro results indicated that PARP1 inhibited *PD-L1* transcription and expression, and could promote the activity of tumor-infiltrating lymphocytes.

**Fig. 3 Knockdown of NPM1 suppresses PD-L1 expression and promotes T cell activity in vivo. a** $0.5 \times 10^6$ 4T1 cells were injected into the mammary fat pads of BALB/c mice. The average tumor volume of each group was depicted over time. **b** The number of lung metastasis nodes of every mouse in each group ($n = 8$) was graphed, and the metastasis rate in teach group was indicated above the bar. **c** Tumor infiltrating lymphocytes (TILs) in the tumors of each group ($n = 5$) were analyzed by flow cytometry. **d** Granzyme B and CD8 in mouse tumor tissues of each group were detected by immunofluorescence (IF). Experiment was repeated in three animal samples with similar results. **e** NPM1 and PD-L1 expression was detected by IHC in mouse tumor tissues of each group. Experiment was repeated in three animal samples with similar results. Data were presented as mean ± s.d. of three independent experiments. **a**–**c** were analyzed by one-way ANOVA + two-side Tukey test. *$P < 0.05$, **$P < 0.01$, ****$P < 0.0001$; ns, not significantly different. Source data are provided as a Source Data file.

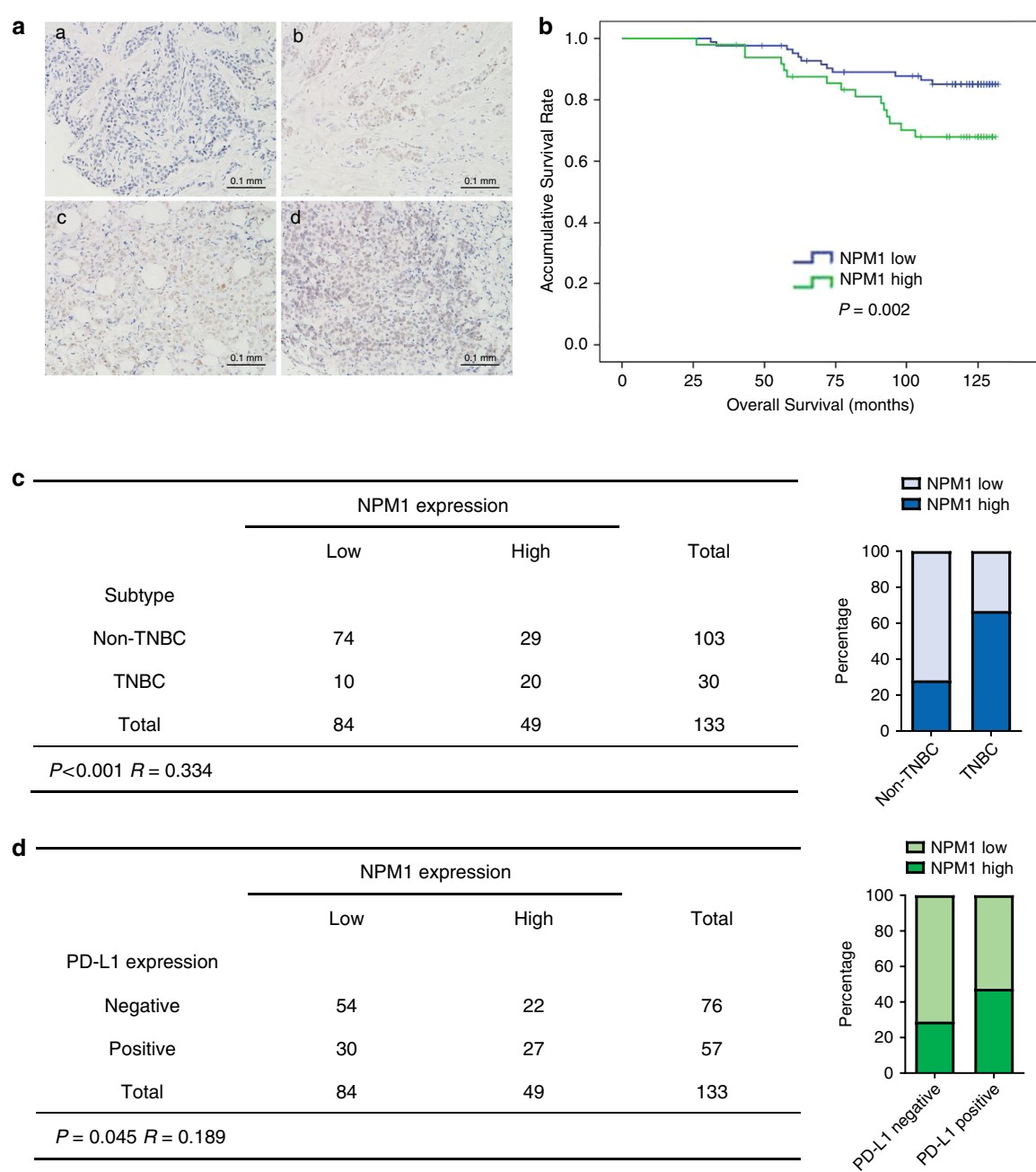

**Fig. 4 NPM1 is a poor prognostic factor and correlated with PD-L1 expression in breast cancer. a** Representative images of IHC of NPM1 in 133 breast cancer tissues (**a**, low expression; **b**–**d** high expression). **b** Kaplan–Meier analysis based on NPM1 expression in 133 breast cancer patients. Data was analyzed by log-rank test. **c** The correlation analysis between NPM1 expression and TNBC status. **d** The correlation analysis between NPM1 expression and PD-L1 expression. Two-side Pearson chi-square analysis was used to determine the correlation.

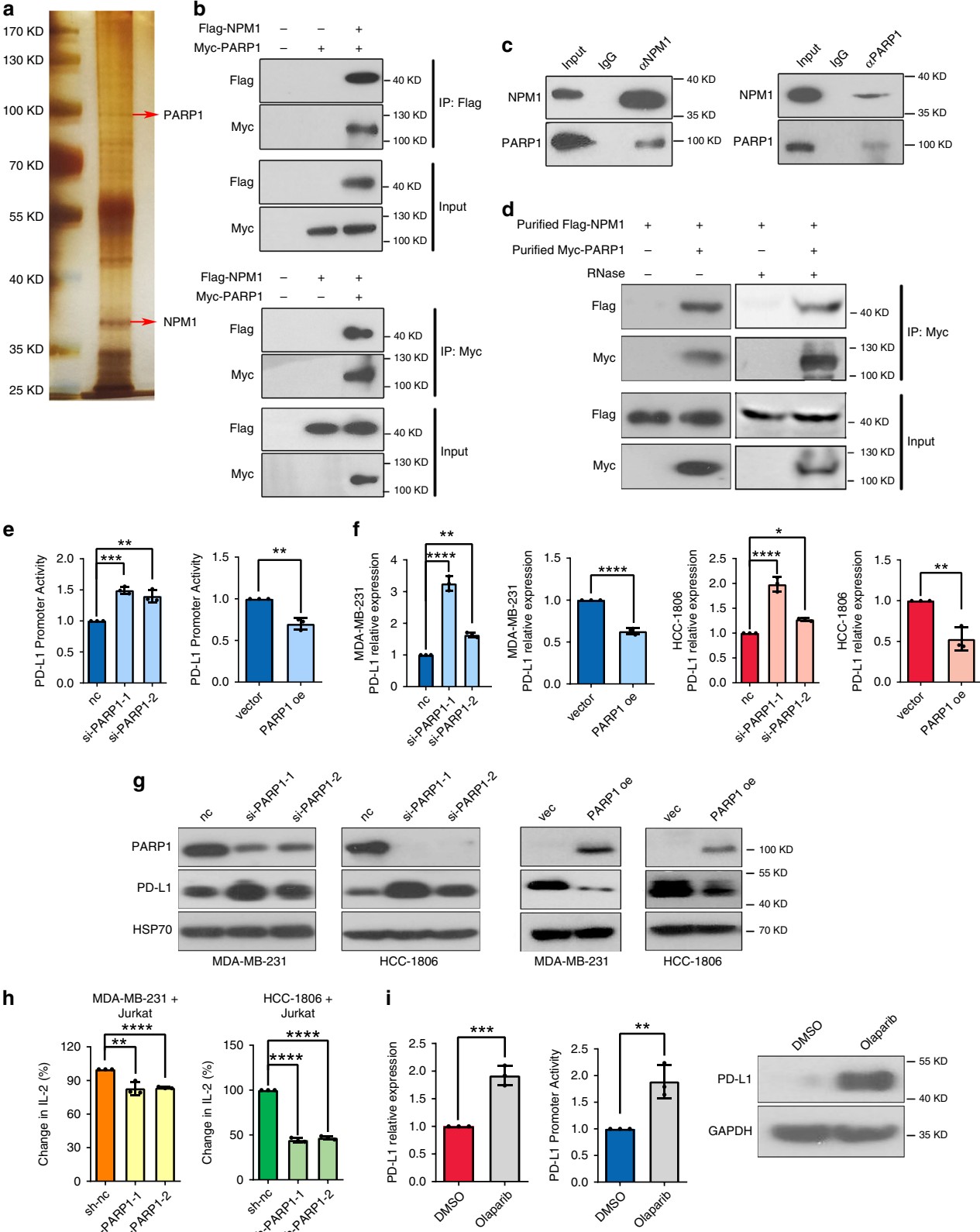

**The regulation of *PD-L1* by PARP1 depends on NPM1**. In light of the observation that PARP1 interacted with NPM1 but the two molecules played opposite roles in the transcriptional regulation of *PD-L1*, we postulated that PARP1 inhibited PD-L1 expression by blocking NPM1 from the promoter of *PD-L1*. To test whether NPM1 mediated the transcriptional regulation of *PD-L1* by

PARP1, NPM1 was knocked down in stable cell lines without PARP1. Noticeably, the increased promoter activity and expression of PD-L1 caused by PARP1 silencing was completely abolished in the absence of NPM1 (Fig. 6a–c). Accordingly, the decreased secretion of IL-2 by Jurkat cells co-cultured with PARP1 deficient cells was fully rescued by NPM1 knockdown

**Fig. 5 PARP1 interacts with NPM1 and inhibits PD-L1 transcription and expression. a** Co-immunoprecipitation (co-IP) assay was conducted with NPM1 antibody in MDA-MB-231 cells. The precipitated proteins were identified by mass spectrometry and the corresponding gene symbols were shown on right. Three biological replicate samples were used. **b** HEK293T cells were transfected with Flag-NPM1 and/or Myc-PARP1. Cell lysates were immunoprecipitated with anti-Flag antibody (top) or anti-Myc antibody (bottom), and then the precipitates were detected with anti-Myc antibody (top) or anti-Flag antibody (bottom). **c** MDA-MB-231 cell lysates were immunoprecipitated with anti-NPM1 (left), anti-PARP1 (right), or control IgG antibody, and then the precipitates were blotted with anti-PARP1 (left) or anti-NPM1 (right) antibody. **d** IP with anti-Myc antibody to detect the interaction between purified Myc-PARP1 and Flag-NPM1 proteins. **e** PD-L1 promoter activity in MDA-MB-231 cells transiently transfected with control siRNA (nc), *PARP1* siRNAs (si-PARP1-1, si-PARP1-2), vector (vec), or PARP1 overexpression plasmid (PARP1 oe) was measured by dual-luciferase assay. **f** PD-L1 mRNA levels in MDA-MB-231 (blue) and HCC1806 (red) cells with transient PARP1 knockdown (left) or overexpression (right) was detected by RT-qPCR. **g** PARP1 and PD-L1 protein expression in MDA-MB-231 and HCC1806 cells with transient PARP1 knockdown or overexpression was detected by Western blot. **h** Jurkat cells were co-cultured with MDA-MB-231 or HCC1806 cells in a 1:4 ratio for 48 h, and secreted IL-2 in the supernatant was detected by ELISA. **i** MDA-MB-231 cells were treated with 25 µM olaparib for 48 h. PD-L1 mRNA expression and promoter activity were respectively measured by RT-qPCR and dual-luciferase assay (left, middle), and PD-L1 protein expression was detected by Western blot (right). Data were presented as mean ± s.d. of three independent experiments. **e** Right panel, **f** 2th, 4th panel and **i** was analyzed by two-side Student's *t*-test, **$P < 0.01$, ***$P < 0.001$, ****$P < 0.0001$. **e** Left panel, **f** 1th, 3rd panel and **h** was analyzed by one-way ANOVA + two-side Dunnett test,*$P < 0.05$, **$P < 0.01$, ***$P < 0.001$, ****$P < 0.0001$. Source data are provided as a Source Data file.

(Fig. 6d). In consistence with these results, the PARP1 inhibitor olaparib failed to upregulate PD-L1 expression without NPM1 (Fig. 6e, f). The results above implicated that the function of PARP1 in *PD-L1* transcriptional regulation depended on NPM1.

Furthermore, ChIP-qPCR assay revealed that NPM1 bound to *PD-L1* promoter while PARP1 did not (Fig. 6g, left). Importantly, the association of NPM1 with *PD-L1* promoter was significantly increased when PARP1 was depleted (Fig. 6g, right), which corroborated our hypothesis that PARP1 interacted with NPM1 to prevent it from binding the promoter of *PD-L1*. To further demarcate the structural domain in NPM1 required for its binding at *PD-L1* promoter, we constructed a plasmid bearing the truncated form of NPM1 that had the last 35 amino acids at its C-terminus deleted[29] (Fig. 6h). This NPM1 Δ35 mutant was still localized to the cell nuclei (Supplementary Fig. 4C). However, NPM1 Δ35 was unable to bind at *PD-L1* promoter (Fig. 6i), and therefore failed to activate the promoter of *PD-L1* (Fig. 6j), as well as the subsequent mRNA transcription and protein expression of the gene (Fig. 6k, l). These results proved that the C-terminal nucleic acid binding domain of NPM1 was essential for its binding at *PD-L1* promoter. Interestingly, the NPM1 Δ35 mutant was also incapable of interacting with PARP1 (Fig. 6m), which suggested that PARP1 bound NPM1 at its C-terminus, thereby blocking its nucleic acid binding domain from associating with the promoter of *PD-L1*.

**Olaparib combined with anti-PD-L1 has better effect**. Given that inhibition of PARP1 with olaparib promoted PD-L1 expression by suppressing NPM1 and PARP1 interaction and enhancing the association of NPM1 at *PD-L1* promoter in human TNBC cells (Supplementary Fig. 5G, H), we speculated that anti-PD-L1 could improve the effect of olaparib on TNBC treatment in vivo. As olaparib was approved by FDA to treat *BRCA1* or *BRCA2* deficient TNBC, we first depleted *BRCA1* by shRNA in mouse 4T1 TNBC cells (Supplementary Fig. 6A), and confirmed that olaparib upregulated PD-L1 expression in these 4T1-BRCA1 (−) cells (Supplementary Fig. 6B).

Next, $0.5 \times 10^6$ 4T1-BRCA1(−) cells were injected into the mammary fat pads of BALB/c mice, and the mice were treated with olaparib ($50 \, \mathrm{mg \, kg^{-1}}$) daily or/and αPD-L1 ($200 \, \mu\mathrm{g \, kg^{-1}}$) once every 5 days (Fig. 7a). The tumor volume of each mouse was measured twice a week, and the results showed that olaparib and αPD-L1 combination therapy most dramatically reduced the tumor size (Fig. 7b and Supplementary Fig. 6C). We also observed that the pulmonary metastasis rate and the number of lung metastatic nodules were the lowest in the combination

therapy group, though not significantly lower than the mono-therapy groups (Supplementary Fig. 6D).

In addition, the CD45+ CD8+ T cell population was relatively low in tumors treated with olaparib alone, but apparently increased in tumors treated with both olaparib and αPD-L1 (Fig. 7c, left and Supplementary Fig. 6E). Similarly, CD107+ and CD69+ T cell infiltration were relatively low in tumors receiving olaparib monotherapy, but upregulated in tumors receiving the combination therapy (Fig. 7c, middle and right and Supplementary Fig. 6F, G). In accordance with our observation in cell lines, IHC staining of the mouse tumor tissues showed that PD-L1 expression was elevated when olaparib was administered (Fig. 7d). We also proved that olaparib had no effect on lymphocytes homing and activation in vitro (Supplementary Fig. 5I, J). Taken together, these results indicated that olaparib increased PD-L1 expression in vivo, whereas application of αPD-L1 in combination with olaparib provided better effects in treating TNBC.

## Discussion
Tumor infiltrating lymphocytes (TILs) have been associated with improved prognosis in many different tumor types[30], as the paucity of tumor T cell infiltration often leads to initial resistance to immunotherapy. Tumors that lack TILs were characterized as "cold tumors", whereas tumors with massive T cell infiltration were defined as "hot tumors". Though each subtype of breast cancer has both "cold" and "hot" tumors, TNBCs often have tumors with >50% lymphocytic infiltration and have higher PD-L1 expression[5,31], which indicate that TNBCs may be the most sensitive to anti-PD-L1 therapy among breast cancer. We analyzed the expression of PD-L1 in 149 breast cancer patients and found PD-L1 positive rate was dramatically higher in TNBC patients and associated with poor prognosis (Fig. 1a–c), which was consistent with other studies[5,9]. In addition, *PD-L1* mRNA expression was also upregulated in TNBC according to TCGA database (Fig. 1d). Therefore, we sought to understand the mechanism that led to the overexpression of PD-L1 in TNBC. Previously, Barrett et al. reported that although TNBCs had higher burden of copy number variants (CNVs), it was not associated with high PD-L1 expression[32]. This made us speculate that there might be additional mechanism regulating the transcription of *PD-L1* in TNBCs. Currently, the majority of studies on *PD-L1* transcriptional activation have focused on pathways induced by inflammatory agents such as IFN-γ and TNF-α[13,33,34]. However, our results indicated that TNBC cell lines had higher intrinsic *PD-L1* mRNA and protein expression without any inducing factor (Fig. 1e, f). Thus, we pursued to discover endogenous transcription regulators of *PD-L1* in TNBCs.

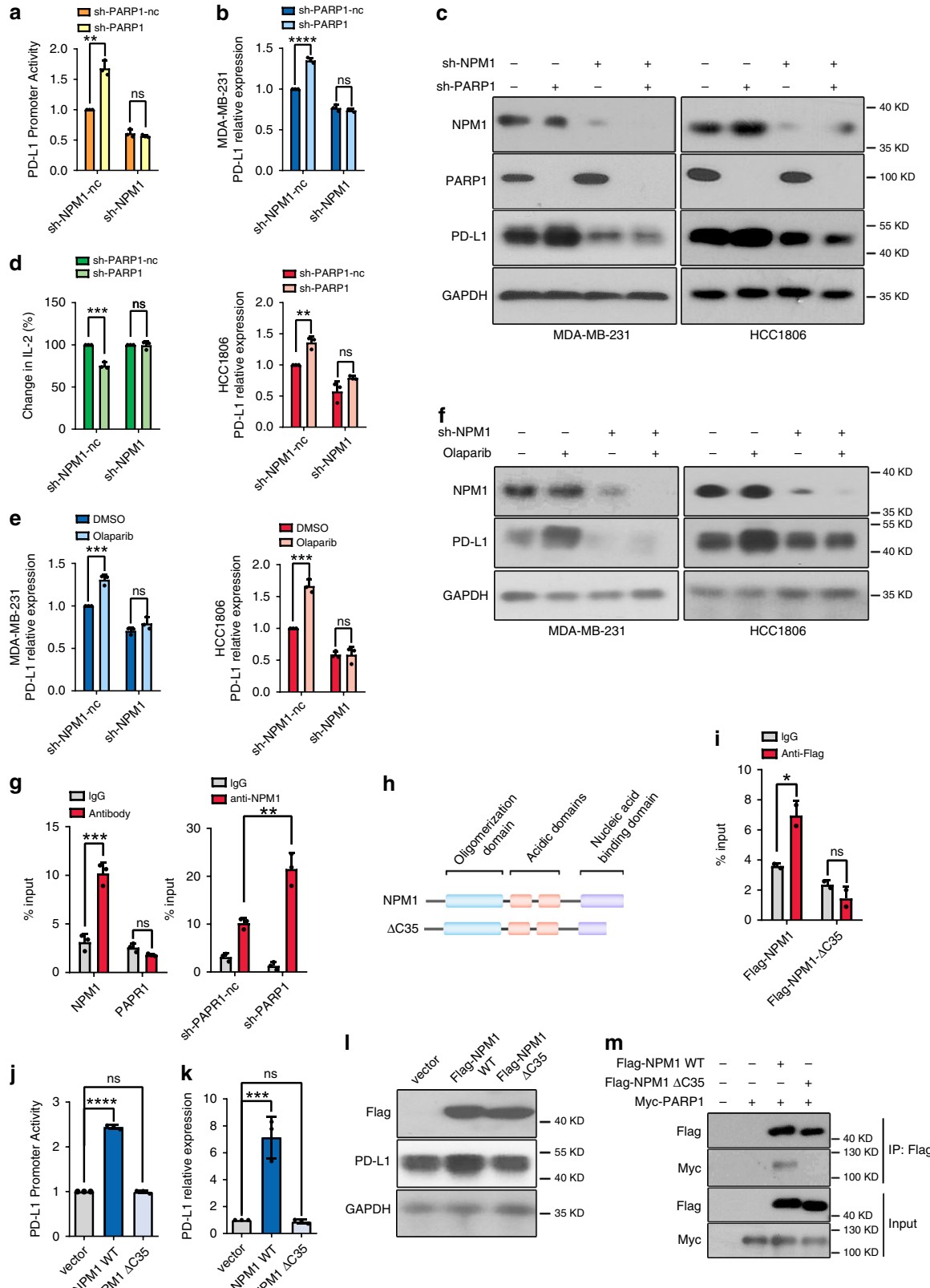

Streptavidin-agarose pull-down assay coupled with mass spectrometry was conducted to identify nuclear proteins specifically bound to a biotin-labeled *PD-L1* promoter probe in TNBC cells, and NPM1 was one of the candidates that intrigued us most. NPM1 is a versatile protein that functions in ribosome biogenesis, mitosis, and DNA replication, transcription and repair[19].

Mutations of the *NPM1* gene is mostly reported in human hematopoietic malignancies, where the N-terminal region of NPM1 often fuses to genes such as anaplastic lymphoma kinase (ALK), retinoic acid receptor α (RARα) or myeloid leukemia factor 1 (MLF1), and NPM1 activates the oncogenic potential of its fused protein partners[35]. In contrast, *NPM1* mutations are

**Fig. 6 The function of PARP1 in *PD-L1* transcriptional regulation depends on NPM1. a** *PD-L1* promoter activity in MDA-MB-231 cells transfected with control shRNA (sh-PARP1-nc or sh-NPM1-nc), *PARP1* shRNA (sh-PARP1), or NPM1 shRNA (sh-NPM1) was measured. **b** PD-L1 mRNA levels in MDA-MB-231 (top) and HCC1806 (bottom) cells stably transfected with control shRNA, *PARP1* shRNA, or *NPM1* shRNA. **c** Protein levels of NPM1, PARP1, and PD-L1 in MDA-MB-231 (left) and HCC1806 (right) cells with stable knockdown of NPM1 or/and PARP1 were detected by Western blot. **d** Jurkat cells were co-cultured with MDA-MB-231 cells with PARP1 or/and NPM1 stably knocked down, and IL-2 was detected by ELISA. **e, f** MDA-MB-231 (left) and HCC1806 (right) cells stably transfected with control or NPM1 shRNA were treated with 25 μM olaparib or vehicle (DMSO). *PD-L1* mRNA (**e**), NPM1 and PD-L1 protein expression (**f**) was detected. **g** ChIP-qPCR of *PD-L1* promoter region was conducted with NPM1, PARP1 or control IgG antibody in MDA-MB-231 cells (left). ChIP-qPCR was conducted with NPM1 or IgG antibody in MDA-MB-231 cells without or with PARP1 stable knockdown (right). **h** The structures of wildtype NPM1 and the ΔC35 mutant of NPM1. **i** ChIP-qPCR of PD-L1 promoter was performed with Flag or IgG antibody in MDA-MB-231 cells transiently transfected with Flag-NPM1 or Flag-NPM1-ΔC35. **j–l** MDA-MB-231 cells were transfected with vector, Flag-NPM1 or Flag-NPM1-ΔC35 plasmid. *PD-L1* promoter activity (**j**), *PD-L1* mRNA (**k**), Flag and PD-L1 protein expression (**l**) was detected. **m** HEK293T cells were transfected with Myc-PARP1 alone, or together with Flag-NPM1 WT or Flag-NPM1 ΔC35. The precipitates were then detected with anti-Myc antibody. Data were presented as mean ± s.d. of three independent experiments. **g, i** were analyzed by two-side Student's *t*-test, *$P < 0.05$, **$P < 0.01$, ***$P < 0.001$; ns, not significantly different. **j, k** were analyzed by one-way ANOVA + two-side Dunnett test, ***$P < 0.001$, ****$P < 0.0001$; ns, not significantly different. **a, b, d, e** were analyzed by ANOVA + two-side Tukey test, **$P < 0.01$, ***$P < 0.001$, ****$P < 0.0001$; ns, not significantly different. Source data are provided as a Source Data file.

scarcely detected in solid tumors[22], and there is limited solid evidence for wildtype *NPM1* acting as a proto-oncogene in vivo, though its overexpression has been detected in gastric, thyroid, liver, and prostate cancers[36–39].

Our experiments in TNBC cells and animal models validated that NPM1 bound to *PD-L1* promoter, increased its activity, and both the mRNA and protein expression of PD-L1 (Figs. 2a–h, 3e and Supplementary Fig. 1B–E), which suggested that NPM1 was an endogenous transcriptional activator of *PD-L1* in TNBC cells that could elevate PD-L1 expression without the induction of exogenous signals. Furthermore, our functional assays showed that knockdown of NPM1 drastically relieved the immunosuppression of T cells, and such an effect was completely reversed by PD-L1 overexpression (Figs. 2i–m, 3a–e and Supplementary Fig. 1F, G), confirming that NPM1 played an immunosuppressive role in TNBC through upregulating PD-L1 transcription.

Previous studies have implied that NPM1 could regulate gene transcription through several mechanisms. Li et al. reported that NPM1 formed a binary complex with c-Myc and bound to the promoters of c-Myc target genes to induce mRNA and rRNA transcription in mouse embryo fibroblasts (MEFs)[40,41]. A recent study demonstrated that c-Myc directly bound to the promoter of PD-L1 and enhanced PD-L1 expression in human melanoma SKMEL28 and human NSCLC H1299 cells[42]. Accordingly, it is possible that NPM1 also interacted with c-Myc in TNBC cells to elevate PD-L1 transcription. Alternatively, NPM1 might serve as a transcription coactivator by binding to NF-κB[43] or other transcription factors to promote PD-L1 expression. Moreover, acetylated NPM1 (Ac-NPM1) was found to have increased affinity to acetylated core histones and facilitate transcription activation through nucleosome disruption[44]. As histone deacetylase inhibitors were shown to up-regulate PD-L1 in melanomas[45], it would be worthwhile to investigate whether NPM1 enhanced *PD-L1* transcription by chromatin remodeling. In addition, NPM1 might act as a transcription factor to regulate PD-L1 expression. It has been reported that NPM1 binds to a G-rich region in DNA with the repetitive sequence "TTAGGG"[46], which is also found in −663 ~ −641 of *PD-L1* promoter region. Our result showed that NPM1 overexpression failed to activate the transcription of PD-L1 controlled by a promoter without the −663 ~ −641 region, indicating this region may play an important role (Supplementary Fig. 5L). However, to rigorously prove that NPM1 is a bona fide transcription factor exceeds the scope of the current study.

Our study has also explored the effect of the NPM1 inhibitor NSC348884 on TNBC. NSC348884 prevents the oligomerization of NPM1, which was required for cell proliferation, but has no effect on NPM1 expression[47]. We noticed that NSC348884 did not change NPM1 protein level or PD-L1 expression in MDA-

MB-231 and 4T1 cells (Supplementary Fig. 7A). Although NSC348884 significantly inhibited tumor growth in vivo (Supplementary Fig. 7B), it neither decreased PD-L1 expression nor increase CD8+ cell infiltration in vivo (Supplementary Fig. 7C, E). These results indicated that the anti-tumor effect of NSC348884 depended on its anti-proliferation effect, and NPM1 regulated *PD-L1* transcription in its monomer form.

We also found that NPM1 had higher expression in TNBCs and was positively correlated with PD-L1 expression (Fig. 4). However, some PD-L1 positive samples did not have high expression of NPM1, which could be due to two reasons. First, PD-L1 expression is not only associated with the total expression of NPM1, but also its active form. It is known that NPM1 can undergo extensive post-translational modifications including phosphorylation[48–50], acetylation[51], ubiquitination[52,53], and SUMOylation[54,55], which are likely to control its stability, localization, interaction with other proteins and the related cellular functions. Hence, we propose that NPM1 monomer may be uniquely modified in TNBC cells to regulate PD-L1 expression. Second, TNBCs have the strongest heterogeneity among all subtypes of breast cancers. Though ER−/PR−/HER-2− breast cancers are all defined as TNBCs, there are different biological behavior, clinical features, therapeutic response and driver genes among them, and the function of NPM1 in regulating PD-L1 expression may be related to the subtypes of TNBCs. These problems will be further studied in our future research.

Noteworthily, we discovered that PARP1 regulated *PD-L1* transcription through its interaction with NPM1. Our data revealed that PARP1 knockdown or inhibition by olaparib enhanced the promoter activity, mRNA transcription and protein expression of *PD-L1*, whereas PARP1 overexpression had the opposite effects (Fig. 5e–h and Supplementary Fig. 5E, F), which suggested that PARP1 suppressed *PD-L1* transcription, opposing the function of NPM1. Surprisingly, this inhibitory effect on *PD-L1* transcription relied on NPM1, and PARP1 knockdown or olaparib failed to promote *PD-L1* transcription in the absence of NPM1 (Fig. 6a–f). These results implicated that PARP1 was unlikely to repress *PD-L1* transcription by competing with NPM1 to bind *PD-L1* promoter. Consistently, our ChIP experiment proved that only NPM1 but not PARP1 associated with the promoter region of *PD-L1* (Fig. 6g). Alternatively, since the C-terminal nucleic acid binding domain of NPM1 was indispensable for its association with *PD-L1* promoter and the consequent activation of PD-L1 expression (Fig. 6h–l), we reasoned that the interaction with PARP1 might mask the C-terminus of NPM1, thus hindering its contact with *PD-L1* promoter. This scenario was corroborated by our finding that the C-terminal nucleic acid binding domain of NPM1 was required for its interaction with

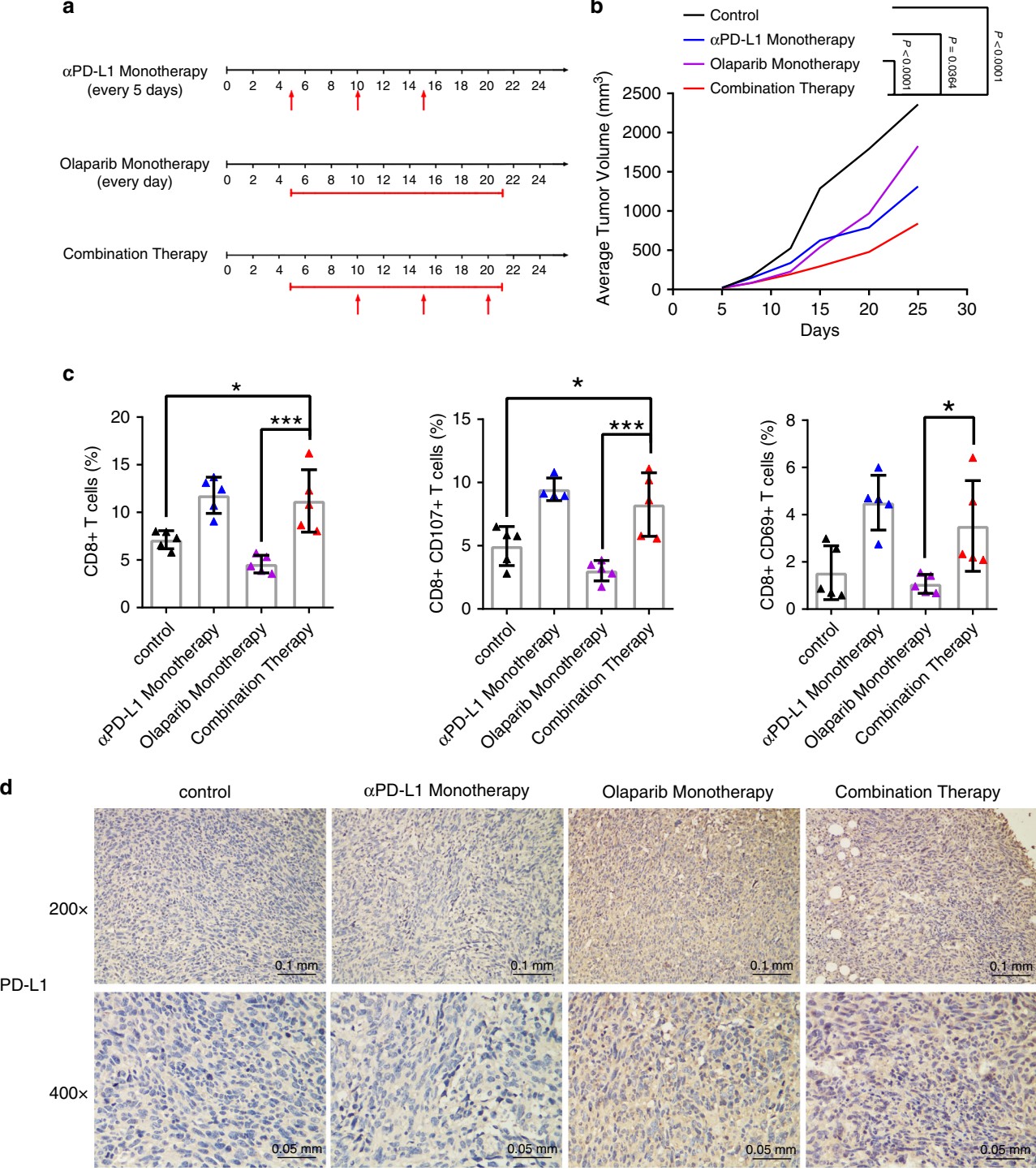

**Fig. 7 Olaparib and anti-PD-L1 combination therapy was superior to monotherapy in vivo. a** Schematic diagram of dosage regimen. Control group was treated with PBS and IgG. αPD-L1 monotherapy group was treated with PBS and αPD-L1 200 μg per mouse. Olaparib monotherapy group was treated with IgG and olaparib 50 mg kg$^{-1}$. Combination therapy group was treated with αPD-L1 200 μg/mouse and olaparib 50 mg kg$^{-1}$. **b** $0.5 \times 10^6$ 4T1-sh-BRCA1 cells were injected into the mammary fat pads of BALB/c mice. The average tumor volume of each group was recorded over time. **c** TILs in the tumors of each group ($n = 5$) were analyzed by flow cytometry. **d** PD-L1 expression was detected by IHC in mouse tumor tissues of each group. Data were presented as mean ± s.d. of three independent experiments. **b**, **c** were analyzed by one-way ANOVA + two-side Tukey test. *$P < 0.05$, ***$P < 0.001$. Source data are provided as a Source Data file.

PARP1 (Fig. 6m). In addition, when PARP1 was knocked down or inhibited by olaparib, NPM1 had enriched binding at *PD-L1* promoter (Fig. 6g and Supplementary Fig. 5H). Therefore, we concluded that PARP1 suppressed PD-L1 expression by preventing NPM1 from binding the promoter of *PD-L1*.

The PARP1 inhibitor olaparib has been approved by FDA to treat TNBC patients with mutations in *BRCA1* or *BRCA2*, and olaparib monotherapy has been proved superior to single-agent chemotherapy, with an increase of median progression free survival from 4.2 to 7.0 months, but no significant improvement in

overall survival[56]. Studies have also indicated that combination therapy of olaparib and platinum chemotherapy could be more beneficial[57,58]. However, chemotherapies are frequently accompanied with side effects that can lead to incompliance. Recently, Jiao et al. reported that olaparib increased PD-L1 protein expression in TNBC, which was consistent with our results[59]. Mechanistically, they proposed that PARP inhibitor inactivated GSK3β to stabilize PD-L1. In our research here, we demonstrated that olaparib abolished the interaction between PARP1 and NPM1, thus exposing the C-terminal nucleic acid binding domain of NPM1 essential for its association with *PD-L1* promoter (Supplementary Fig. 5G, H). Our finding is not mutually exclusive with that by Jiao et al., but has elucidated an additional layer of regulation of PD-L1 expression by PARP inhibitor. Currently, the immunoregulation activity of olaparib is still controversial. On the one hand, as is shown by us here and others, olaparib enhances PD-L1 expression, while on the other hand, olaparib has also been reported to stimulate CD8+ T cell infiltration by upregulating the STING pathway[60]. So far, there has been no evidence that olaparib will promote tumor growth, increase metastatic rate, or induce irreversible immunosuppression by upregulating PD-L1 expression. It is conceivable that the dominant effect of olaparib is its cytotoxicity, and the immunosuppressive activity caused by increased PD-L1 expression is probably a critical factor that weakens the anti-tumor effects of olaparib in vivo. Our experiment in mouse model showed that the combination therapy with olaparib and anti-PD-L1 had better effect than monotherapy in BRCA1 deficient TNBC xenografts, suggesting that the combination regimen might not only improve the curative effect of olaparib, but also have lower toxicity than the combination therapies using olaparib with chemotherapeutic agents.

In summary, we have identified NPM1 as a transcriptional regulator of *PD-L1*, which particularly promotes PD-L1 expression in TNBC, and is correlated with poor prognosis in breast cancer. Therefore, NPM1 may be a target for TNBC treatment and potentially other cancer types as well. In addition, an inhibitory role of PARP1 in *PD-L1* transcription mediated by NPM1 has been revealed in this research. Importantly, this regulatory mechanism provides solid foundation for combination therapy with anti-PD-L1 and PARP inhibitor in TNBC.

## Methods

**Cell culture and reagents**. All cell lines used in this research were obtained from American Type Culture Collection (ATCC, Manassas, VA, USA). MDA-MB-231, BT-20, HS578T, SKBR3, HEK293T, and B16 were grown in DMEM (Invitrogen, Carlsbad, CA, USA) supplemented with 10% fetal bovine serum (Invitrogen), 100 U mL$^{-1}$ penicillin and 100 μg mL$^{-1}$ streptomycin (Invitrogen). HCC1806, HCC1937, MCF-7, T47D, 4T1, and Jurkat were cultured with RIPM 1640 (Invitrogen) supplemented with 10% fetal bovine serum (Invitrogen), 100 U mL$^{-1}$ penicillin and 100 μg mL$^{-1}$ streptomycin (Invitrogen). All cells were cultured in a humidified atmosphere of 5% CO$_2$ at 37 °C.

**Plasmids and small interfering RNA transfection**. The Flag-tagged NPM1, Myc-tagged PARP1 and Flag-tagged NPM1 Δ35 fragments were cloned into the pSIN-EF2-puro vector, respectively. The different fragments of the promoter region of PD-L1 were cloned into the pGL3-basic vector. All plasmids were transfected with Lipofectamine@3000 transfection reagent (L3000-150; Invitrogen). The sequences of small interfering RNA were listed below: si-NPM1-1 5′-GGAAUGUUAUGAU AGGACATT-3′, si-NPM1-2 5′-AGGTGGTAGCAAGGTTCCA-3′, si-PARP1-1 5′-GAGUAUGCCAAGUCCAACATT-3′, si-PARP1-2 5′-GGAACAAGGAUGAA GUGAATT-3′. All si-RNAs were tranfected with Lipofectamine@ RNAIMAX transfection reagent (13778-150; invitrogen).

**Stable transfection using lentiviral infection**. The PLKO.1-puro vector was used to clone the sh-RNAs targeting *NPM1*, *PARP1*, and *BRCA1*. The sequences of sh-NPM1 and sh-PARP1 were the same as their si-RNAs. The sequence of sh-BRCA1 was 5′-CCGGCCTCACTTTAACTGACGCAATCTCGAGATTGCGTCAGTTAA AGGAGGTTTTTG-3′, which targeted mouse BRCA1. Human PD-L1 and mouse PD-L1 were cloned into the pSIN-EF2-blasticidin vector. The plasmids were transfected into HEK293T cells and the supernatant which contained the virus was collected at 48 and 72 h. The virus was then concentrated and transfected in breast cancer cells with polybrene (sc-134220, Santa Cruz). The transfected cells were selected by puromycin or blasticidin S for at least 1 week.

**Western blot analysis**. Cells were harvested and proteins were extracted with RIPA lysis buffer. The following antibodies were used: NPM1 (FC-61991; Invitrogen, Carlsbad, CA, USA; 1:1000), PD-L1 (GTX104763; GeneTex, Irvine, CA, USA; 1:4000), PD-L1 (GTX31308; GeneTex; 1:1000), PD-L1 (13684; Cell Signaling Technology, Danvers, MA, USA; 1:1000), PARP1 (9532; Cell Signaling Technology; 1:1000), Flag (14793; Cell Signaling Technology; 1:1000), Myc (16286-1-AP; Proteintech, Rosemont, IL, USA; 1:1000), HSP70 (46477; Cell Signaling Technology; 1:1000), GAPDH (10494-1-AP; Proteintech; 1:10,000). Cytokine IFN-γ (EST-IFg-0100; Stemimmune LLC, Richmond, CA, USA) was used to treat cells at the concentration of 25 μg mL$^{-1}$.

**Cell surface PD-L1 expression analysis**. Cells were digested to single cells and washed by cold PBS for three times. Cells were resuspended with 100 μL PBS and incubated with CD274-PE (12-5983-42; eBioscience; 1:20) or Mouse IgG1 kappa Isotype Control-PE (12-4714-82; eBioscience; 1:20) for 15 min on ice. Then cells were washed with cold PBS and detected by flow cytometry.

**RNA extraction and RT-qPCR assay**. Total RNA was isolated by RaPure Total RNA Micro Kit (R4012; Magen, Guangzhou, GD, China). First stand c-DNA was synthesized using HiScript II Q RT SuperMix for qPCR kit (R223-01; Vazyme, Piscataway, NJ, USA). ChamQ SYBR qPCR Green Master Mix (Q311-02; Vazyme) was used to conduct qPCR assay. The primers used were listed below: *NPM1* forward, 5′-GGAGGTGGTAGCAAGGTTCC-3′, *NPM1* reverse, 5′-TTCACTGG CGCTTTTTCTTCA-3′; *PD-L1* forward, 5′-TGGCATTTGCTGAACGCATTT-3′, *PD-L1* reverse, 5′-TGCAGCCAGGTCTAATTGTTTT-3′; *PARP1* forward, 5′-TG GAAAAGTCCCACACTGGTA-3′, *PARP1* reverse, 5′-AAGCTCAGAGAACCC ATCCAC-3′; *GAPDH* forward, 5′-ATCACCATCTTCCAGGAGCGA-3′, *GAPDH* reverse, 5′-CCTTCTCCATGGTGGTGAAGAC-3′ (Supplementary Table 5).

**Dual luciferase reporter assay**. MDA-MB-231 cells were seeded in 24-well plates and transfected with 0.5 μg/well luciferase reporter plasmids. To normalize the transfection efficiency, the cells were co-transfected with 10 ng of pRL-CMV (Renilla luciferase). Forty-eight hours after transfection, the luciferase activity was detected using Dual-Luciferase Reporter Assay System Kit (E1910; Promega, Madison, WI, USA) according to the manufacturer's instruction.

**Streptavidin-agarose pull-down assay**. The *PD-L1* promoter sequence was obtained from the UCSC database, and a 666 bp (−24 to −690 of the PD-L1 promoter region) biotin-labeled double-stranded DNA probe was synthesized by PCR with a pair of biotin-labeled primers (Foward 5′-GGCTGCGGAAGCCTA TTCTA-3′, reverse 5′-ACCTCTGCCCAAGGCAGCAA-3′). Nuclear proteins were extracted from both TNBC cell lines and non-TNBC cell lines. The biotin-labeled PD-L1 promoter probe (4 μg) was added to the nuclear lysate (400 μg, 1 mL) with 40 μL streptavidin-agarose beads (S1638; Sigma, St. Louis, MO). After incubation overnight at 4 °C with rotation, the beads were collected by centrifugation and washed twice with lysis buffer. Afterward, the samples were eluted with 20 μL SDS loading buffer by boiling at 100 °C for 10 min. The pull-down proteins were separated by polyacrylamide gel electrophoresis and detected by sliver staining.

**Mass spectrometry analysis**. The in-gel digestion was carried out as described by Katayama et al.[61]. Next, samples were re-suspended with Nano-RPLC buffer A. The online Nano-RPLC was employed on the Eksigent nanoLC-Ultra™ 2D System (AB SCIEX). The samples were loaded on C18 nanoLC trap column (100 μm × 3 cm, C18, 3 μm, 150 Å) and washed by Nano-RPLC Buffer A (0.1% FA, 2% ACN) at 2 μL min$^{-1}$ for 10 mins. An elution gradient of 5–35% acetonitrile (0.1% formic acid) in 90 min gradient was used on an analytical ChromXP C18 column (75 μm × 15 cm, C18, 3 μm 120 Å) with spray tip. Data acquisition was performed with a Triple TOF 5600 System (AB SCIEX, USA) fitted with a Nanospray III source (AB SCIEX, USA) and a pulled quartz tip as the emitter (New Objectives, USA). Based on combined MS and MS/MS spectra, proteins were successfully identified based on 95% or higher confidence interval of their scores in the MASCOT V2.3 search engine (Matrix Science Ltd., London, UK).

**ChIP-qPCR assay**. The ChIP assay was conducted using SimpleChIP@ Enzymatic Chromatin IP kit (9002S, Cell Signaling Technology) following the manufacturer's instruction. Briefly, cells were cultured to about 1 × 10$^7$ and cross-linked by 1% formaldehyde. Samples were then harvested and digested chromatin with micrococcal nuclease. Next, several pulses were used to break nuclear membrane. DNA fragment length was checked to be between 150–900 bp. Chromatin was immunoprecipitated by either control IgG or NPM1 (FC-61991; Invitrogen; 2 μg), Flag (14793; Cell Signaling Technology; 4 μg) or Myc (2267; Cell Signaling Technology; 4 μg) primary anti-body. After washes and reverse cross-link. The eluted DNAs were quantified by qPCR. The primer sequences were listed below: *PD-L1*

(−247 ~ −24 bp), forward 5′-CTTCGAAACTCTTCCCGGTG-3′, reverse 5′-ACC TCTGCCCAAGGCAGCAA-3′; PD-L1 (−469 ~ −247 bp), forward 5′-AAACCAA AGCCATATGGGTC-3′, reverse 5′-AGCCAACATCTGAACGCACC-3′; PD-L1 (−690 ~ −469 bp), forward 5′-TAGAATAGGCTTCCGCAGCC-3′, reverse 5′-CT AGAAAGTAGGTGTGTGTG-3′ (Supplementary Table 5).

**Protein purification.** HEK293T cells were transfected with Flag-NPM1 or Myc-PARP1 plasmids respectively and cultured for 48 h. The cells were harvested and lysed. Anti-Flag magnetic beads (B26010, Bimake, Houston, TX) or anti-Myc magnetic beads (B26301, Bimake) were added to corresponding cell lysates and incubated overnight at 4 °C. The beads were washed with RIPA for five times. Next, 100 µg mL⁻¹ 3 × Flag peptides (200 µL, A6001, APExBio, Houston, TX, USA) or 200 µg mL⁻¹ c-Myc peptides (200 µL, A6003, APExBio) were added to the beads to elute the purified proteins at 4 °C for 2 h. The elution step was repeated once. Four hundred microliter purified protein solution was collected, and 20 µL purified protein sample was used for gel electrophoresis and Coomassie Brilliant Blue staining.

**Co-Immunoprecipitation (Co-IP).** For co-IP, the cells were first transfected with plasmids bearing Flag and/or Myc tagged protein genes. Cells were then lysed and the supernatants were incubated with anti-Flag-agarose beads (A4596; Sigma, St. Louis, MO, USA; 20 µL) or anti-Myc-agarose beads (ab1253, Abcam; 20 µL) overnight at 4 °C, and the precipitates were washed five times with RIPA. To investigate the interaction between endogenous NPM1 and PARP1, the supernatants of cell lysates were first incubated with an NPM1 antibody for 2 h at 4 °C. Protein A/G-agarose (sc-2003; Santa Cruz, CA, USA; 20 µL) was then added and incubated overnight. The precipitates were washed five times with RIPA and analyzed by Western blotting. For IP in purified proteins, purified Flag-NPM1 protein was mixed with purified Myc-PARP1 protein. Anti-Myc magnetic beads were added to the mixture and cultured at 4 °C overnight. The precipitates were then washed five times with RIPA and analyzed by Western blot.

**Confocal immunofluorescence assay.** IF assay was performed in both cells and tumor tissues from mice. For cell lines, cells were fixed with 4% paraformaldehyde for 15 min, permeabilized with 0.5% Triton-X for 5 min and then blocked with 5% bovine serum albumin (BSA) for 30 min. The primary antibodies against NPM1 (FC-6199; Invitrogen; 1:50), PARP1 (9532; Cell Signaling Technology; 1:50) and Flag (14793; Cell Signaling Technology; 1:50) were diluted in 1% BSA and incubated at 4 °C over night. Then, the secondary antibodies, Dylight 549 (A23320; Abbkine, Wuhan, HB, China; 1:200) and Dylight 488 (A23210; Abbkine; 1:200) were added to the samples at dilution of 1:200 and incubated at room temperature for 30 min. One percentage BSA in PBS was included as a negative control when cells were incubated with primary antibodies. The cell nuclei were stained with 0.5 µg mL⁻¹ of 4′,6-diamidino-2-henylindole (DAPI).

For tumor tissues, samples were soaked in dimethyl benzene for 15 min twice and then washed with ethyl alcohol according to concentration gradient. Antigen retrieval was conducted with EDTA (ph = 8.0) and samples were blocked with 10% goat serum. The primary antibodies of eFlour615-CD8 (42-0081-82; eBioscience, Carlsbad, CA, USA; 1:50) and Granzym B (ab4059; Abcam, Cambridge, MA, USA; 1:50) were added to samples and incubated at 4 °C over night. Confocal immunofluorescence results were collected by FV10-ASW 1.7 Viewer.

**Jurkat co-culture and IL-2 ELISA assay.** MDA-MB-231 and HCC1806 cells were treated with IFN-γ for 24 h. Jurkat cells were stimulated by 50 ng L⁻¹ PMA and 1 µg mL⁻¹ Ionomycin for 24 h. Subsequently, 0.5 × 10⁴ MDA-MB-231 cells or 1 × 10⁴ HCC1806 cells were seeded in 96-well plates. After the cells adhered, the supernatants were discarded, and Jurkat cells were added to MDA-MB-231 or HCC1806 cells at a ratio of 4:1 in 200 µL media. The supernatants were collected after 48 h and examined by Human IL-2 Valukine ELISA kit (VAL110; Novus, Littleton, CO, USA) according to the manufacturer's instruction, and the results were analyzed by ELISACalc V0.1.

**Primary T cell co-culture and IFN-γ production assay.** MDA-MB-231 and HCC1806 cells were treated with IFN-γ for 24 h. Primary T cells were stimulated by 50 ng L⁻¹ PMA and 1 µg mL⁻¹ Ionomycin for 24 h. Next, 2 × 10⁴ MDA-MB-231 cells or HCC1806 cells were seeded in 24-well plates. After the cells adhered, the supernatants were discarded. Primary T cells were added to MDA-MB-231 or HCC 1806 cells at a ratio of 30:1 in 500 µL media. Primary T cells were collected after 24 h and treated with Protein Transport Inhibitor Cocktail (00-4980-93; eBioscience) for 6 h. For B16-OVA and OT-I mouse splenic lymphocytes co-culture assay, OT-I mouse splenic lymphocytes were separated and stimulated with 1 µg mL⁻¹ OVA peptide (vac-sin, InvivoGen, San Diego, CA, USA) for 24 h. The lymphocytes were then co-cultured with B16-OVA cells in a 6:1 ratio for 6 h and treated with Protein Transport Inhibitor Cocktail. Afterward, primary T cells or OT-I mouse splenic lymphocytes were fixed and permeabilized with Intracellular Fixation & Permeabilization Buffer Set (88-8824-00; eBioscience) according to the manufacturer's instruction. Samples were then incubated with IFN-γ antibody (17-7319-41; eBioscienc; 1:20) for 30 min and analyzed by flow cytometry and data was collected by CytExpert (2.2.0.97).

**Transwell assay for lymphocytes.** Five micrometer transwell chambers were pre-coated with 20 µg mL⁻¹ ICAM-1 (10346-H08H, SinoBiological, Wayne, PA, USA). Peripheral blood mononuclear cells (PBMCs) from health people were stimulated by 50 µg L⁻¹ PMA and 1 µg mL⁻¹ Ionomycin for 24 h and then 3 × 10⁵ PBMCs were added to the upper chamber. 250 ng mL⁻¹ CXCL10 (10768-HNAE, Sino-Biological) was added to the lower chamber. After 3 h, the migrated cells were counted manually under microscope.

**PAPR1 enzymatic activity measurement.** 20 µg mL⁻¹ cisplatin was used to induce the DNA double-strand damage in 4T1 cells. The cells were then treated with DMSO or 25 µM olaparib for 48 h. Nuclear proteins were extracted and quantified. PARP1 enzymatic activity was measured by the PARP1 activity quantification kit (GMS50116.1, GENMAD, Shanghai, CHN) according to the manufacturer's instruction.

**Animals and treatment.** Female BALB/c mice aged 5–6 weeks were purchased from Beijing Vital River Laboratory Animal Technology and quarantined for one week before use. Animal care and experiments involved in this study were performed in accordance with Accreditation of Laboratory Animal Care International guidelines. Animal experiment protocols were approved by the guidelines established by the Animal Care Committee of Sun Yat-sen University Cancer Center. 0.5 × 10⁶ 4T1 cells were suspended in 40 µL of normal saline (NS) and injected to the mammary fat pads through operation. The tumor sizes were measured twice a week. Tumor volume (TV) was calculated as TV (mm³) = π/6 × length × width². Animals were sacrificed due to progressive disease if tumor burden was greater than 2500 mm³ according to the Animal Care Guidelines of our institute. The size of a few tumors were smaller than 2500 m³ at the penultimate measurement, but these tumors rapidly increased beyond our expectation and exceed 2500 m³ at the last time point. These mice were humanely sacrificed after measurement. For experimental accuracy, these data were also included. For in vivo combination therapy, the control group was treated with PBS and IgG. αPD-L1 monotherapy group was treated with PBS and αPD-L1 (BE0101; BioXcell, West Lebanon, NH, USA) 200 µg/mouse. Olaparib monotherapy group was treated with IgG and olaparib (S1060; Selleck Chemicals, Houston, TX, USA) 50 mg kg⁻¹. Combination therapy group was treated with αPD-L1 200 µg/mouse and olaparib 50 mg kg⁻¹. For in vivo experiment of NSC348884, 1 × 10⁶ 4T1 cells were subcutaneously injected in female BALB/c mice. The mice were treated with 12.5 mg kg⁻¹ NSC348884 (S8149, Selleck Chemicals) or vehicle twice a week for 3 weeks. The tumor volume was measured every 5 days.

**Tumor tissue digestion and flow cytometry analysis.** The mammary tumor nodes of mice were excised and divided into two parts for histopathological analysis and flow cytometry analysis respectively. The tumor tissues for flow cytometry were cut into small pieces and digested with 1 mg mL⁻¹ collagenase type IV (C5138; Sigma) and 0.6 ku mL⁻¹ DNAse (D5025; Sigma) for 2.5 h. Samples were then filtrated to single-cell suspension. Cells were stained with CD45-FITC (11-0451-82; eBioscience; 1:20), CD8-PE (12-0081-82; eBioscience; 1:20), CD107-APC (MA5-28671; eBioscience; 1:20) and CD69-PerCP-Cyanine5.5 (45-0691-82; eBioscience; 1:20). Subsequently, cells were analyzed by flow cytometry. The results were analyzed by FlowJo X.

**Human tissue specimens.** A total of 149 paraffin-embedded primary specimens were obtained from the recruited breast cancer patients. The patients were diagnosed according to their clinicopathologic characteristics from 2000 to 2012. The median age of the patients at diagnosis was 47.0 years (ranging from 21 to 84 years). The patients were staged according to the Union for International Cancer Control TNM staging system. Resected specimens were macroscopically examined to determine the location and size of a tumor, and specimens for histology were fixed in 10% (v/v) formalin and processed for paraffin embedding. Informed consent was obtained from all patients and approved by the research medical ethics committee of Sun Yat-sen University. Besides, a set of tissue microarray including 140 breast cancer cases were bought from Outdo Biotech company (HBreD140Su04). The ages of these patients ranged from 29 to 87. These tissues were all obtained at their first operation with no previous treatment. The therapeutic regimens of these patients were not provided.

**Histopathology.** Immunohistochemical (IHC) staining was performed on 3-µm sections. The primary antibodies against NPM1 or PD-L1 were diluted 1:100, and then incubated at 4 °C overnight. After three washes with PBS, the tissue slides were treated with a non-biotin horseradish peroxidase detection system according to manufacturer's instructions (K500112; Dako, Santa clara, CA, USA). The IHC results were evaluated by two independent pathologists blinded of clinical information. PD-L1 positive was defined as positive cells >1%. NPM1 expression was scored by the staining intensity and percentage of positive cells. The intensity was classified into four scores: "0" for no brown particle staining, "1" for light brown particles, "2" for moderate brown particles, and "3" for dark brown particles. The percentage of positive cells was also divided into four scores: "0" for <10% positive cells, "1" for 10–40% positive cells, "2" for 40–70% positive cells, and "3" for ≥ 70%

positive cells. The two scores were multiplied and used to determine high (score ≥ 3) or low (score < 3) expression of NPM1.

**Database.** PD-L1 promoter sequence was obtained from UCSC database [https://genome.ucsc.edu]. We analyzed PD-L1 expression in breast cancer using TCGA data in Oncomine [http://www.oncomine.org]. Kaplan–Meier analysis of NPM1 in breast cancer was conducted in Kaplan–Meier Plotter [http://www.kmplot.com]. Transcription regulator prediction of PD-L1 was conducted in GCBI [https://www.gcbi.com.cn].

**Statistics and reproducibility.** All data were analyzed and graphed using GraphPad Prism 6.0 or SPSS (version 16.0, SPSS Inc.). The experimental data were presented as mean ± s.d. of three independent trials. Two group comparison was analyzed by two-side Student's $t$-test, and multiple group comparison was analyzed by one-way ANOVA + two-side Dunnett test (when each group compared with a control group) or one-way ANOVA + two-side Tukey test (when each group compared with every other group). The correlation analysis was conducted by Pearson Chi-Square. The survival analysis was performed by Kaplan–Meier Curve and COX regression analysis. $P < 0.05$ was considered statistically significant. Western blots were repeated three times independently with similar results.

**Reporting summary.** Further information on research design is available in the Nature Research Reporting Summary linked to this article.

## Data availability
All data supporting the findings of this study are available with the article and Supplementary information or from the corresponding author upon reasonable request. The source data underlying Figs. 1e, f, 2b, 2e–m, 3a–c, 5b–i, 6a–g, 6i–m, and 7b, c and Supplementary Figs. 1C, D, 1F, 2A–E, 4A, 5E, 5G–L, 6A–D, and 7A–D are provided as a Source Data file. Mass spectrometry data is provided in Supplementary Data 1.

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

## Acknowledgements

This work was supported by grants from the National Natural Science Foundation of China (81772925, 81972569, 81672665, 81672273, 81702761, 81972623, 81702890, and 81703060), the Sci-Tech Project Foundation of Guangzhou City (201607020038), the Guangdong Esophageal Cancer Institute Science and Technology Program (M201802), and the Natural Science Foundation of Guangdong Province (2017A030313615).

## Author contributions

Study concepts: G.Q., W.D. Study design: G.Q., X.W., S.Y., Y.L., M.C. Literature research: G.Q., X.W., T.Q., Y.L. Experimental studies: G.Q., X.W., S.Y., Q.L., C.Z., D.S., J.L., K.Z., Q.Z. Data acquisition: S.W., H.H., Y.T., F.X., J.L. Data analysis/interpretation: G.Q., X.W., S.Y. Statistical analysis: G.Q. Manuscript preparation: G.Q., X.W. Manuscript definition of intellectual content: G.Q., T.K., P.L., W.D. Manuscript editing: G.Q., M.C., W.D. Manuscript revision: G.Q., M.C., X.W., W.D. Manuscript final version approval: W.D.

## Competing interests

The authors declare no competing interests.
