## [Peer Review File · Nature Communications]

Reviewers' comments:

Reviewer #1 (Remarks to the Author):

1. The figures have too many panels that can be placed in the supplement and only a summary panel provided in the main figure.
2. In supplementary table 1, it is unclear what the significant P values are referring to. For example, when considering tumor size, what is driving the significant p-Value? Is it the fact that a tumor is less likely to be PDL1 negative if it is <20 mm? This needs to be defined for all the characteristics listed where there is significance.
3. In Supplementary table 1, what is the median age for the PDL1 positive group?
4. Can the Kaplan Meir survival curves be broken up by stage and PDL1 status?
5. In Figure 2 D, were other proteins identified within the "NPM1" band that was cut out for mass spectrometry?
6. Representative mass spectrometry data must be provided in the supplement.
7. Figure 2 has too many panels that should be placed in the supplement. Some of the data is redundant. It should be included, but not in the primary figure.
8. What was the effect of sh-NPM1 on the surface expression of PD-L1 in the other cell lines? All this data, including Fig.2J, can be shown in the supplement.
9. Why do the authors only focus on a couple of cell lines for further studying NPM1 and PDL1, when they initially introduce an number of cell lines? It is perfectly acceptable to study only 1 or 2 of the cell lines, but a rationale needs to be provided. Or alternatively, if the authors performed the experiments in all the TNBC cell lines, then they should show these data in the supplement.
10. The use of Jurkat cells as the effector cells to show the effects of NPM1/PDL1/PARP on immune cell functions needs to be justified. Ideally, the authors should expand T cells (from healthy donor peripheral blood) that react with the tumor cell line using a known tumor antigen for the cognate cell line or using tumor cell lysates. Is there data to show that Jurkat cells recognize MDA-MB-231 and HCC1806? If so, references in well-respected journals should be provided to support the use of this model.
11. In figure 3, the authors need to state what PDL1-vec means. Is this the negative control vector for PDL1?
12. Figure 3: too many panels. Much of the data can be included in the supplement. For example, Figure 3 C is the summary figure and should be included, while 3B should be in the supplement.
13. The authors should discuss briefly cold vs. hot tumors as they relate to PDL1 expression and TNBC.
14. Did the authors look at the expression of cytokines and chemokines in the tumor tissue that may have contributed to the increase in T cell infiltration in the in vivo model? If so, please show data. If not, please discuss why you think the decrease in NPM1 and PDL1 contributed to the increase in TIL.
15. The scientific studies showing that NPM1 regulates PDL1 are convincing. However, the claim that NPM1 is an independent poor clinical prognosticator in TNBC is inappropriate to state without performing multivariate analyses to demonstrate this fact. The authors should CLEARLY state that their results are based on univariate, possibly biased analyses and MUST be confirmed in larger cohorts with appropriate statistical analyses.
16. The patient characteristics of the cohort that was used in Fig. 4 need to be published in the supplemental material.
17. Figures 4C and D: Which comparisons are driving the statistical significance?
18. There is no information in the methods on mass spectrometry. This needs to be added.
19. Figure 5A: there is no band adjacent the arrow indicating PARP.
20. Figure 5 G Figure legend states NPM1 knockdown or overexpression. Should this be PARP1 (not NPM1)?
21. A reference needs to be added showing that olaparib inhibits MOUSE PARP1.
22. Fig. 7: the authors should demonstrate the effects of olaparib on T cell functions, including T cell homing and activation, and expression of conventional T cell surface markers. This can be done in in vitro assays.

Minor comment:

1. The word "Besides" should be replaced with a more appropriate synonym throughout the manuscript.
2. Need to define "oe" and "nc" in the figure 2 legend.
3. Some of the labels for many figures, (example Figure 5G) are cut off.
4. Figure 5E in the HC1806 data, PARP is spelled as PAPP.
5. Figure 5G PARP is mis-spelled.
6. Figure 6: some of the labels are cut off.

Reviewer #2 (Remarks to the Author):

In the manuscript entitled "NPM1 upregulates the transcription of PD-L1 and suppresses T-Cell activity in triple negative breast cancer," the authors explore the functions of nucleophosmin (NPM1) as a novel regulator of programmed cell death protein-1 (PD-L1) in triple-negative breast cancer (TNBC). They first show that PD-L1 expression is higher in TNBC than other breast cancer subtypes, then map out the region of the PD-L1 promoter responsible for gene transcription, and then identify NPM1 as a protein that physical interacts with this region of the PD-L1 promoter. They then show that NPM1 binds to PD-L1 promoter which in turn results in increased expression of PD-L1 in TNBC. Their functional assays show that NPM1 knockdown promotes T cell activation, while the overexpression of NPM1 has the opposite effect. Interestingly, the authors find that PARP-1 inhibits the expression of PD-L1. Mechanistically, they show that PARP-1 inhibits the expression of NPM1 through its interaction with NPM1. In line with this observation, they show that treatment of TNBC cells with olaparib suppresses the activity of tumor infiltrating lymphocytes (TILs) and sensitizes TNBC to anti PD-L1 therapy in vivo. Taken together, the results presented by the authors indicate that NPM1 is a novel regulator of PD-L1 and thus, NPM1 can be a potential therapeutic target in TNBC.

The findings presented by authors are novel, intriguing and relatively comprehensive. The manuscript is well written. I have only a few comments regarding changes that would improve this study; they are summarized below:

1. NPM1 is an abundant protein, not tissue specific, and is known to interact with many different proteins. Can the authors show a direct interaction of NPM1 and PARP-1 in a cell-free system?
2. The authors show that PARP-1 interacts with NPM1 and suppresses the expression of PD-L1. Have they tested if PARP inhibitors disrupt the interaction of NPM1 and PARP-1?
3. Do NPM1 inhibitors, such as NSC34884, reduce TNBC xenograft growth and increase the infiltration/activity of CD8+ T cells?
4. The authors show that NPM1 expression is positively correlated with PD-L1 expression in approximately 57% of TNBC samples. Could they comment on the possible reasons for high PD-L1 expression in samples that do not have high expression of NPM1?

While the study could be strengthened by addressing the comments above, this is overall a nice study which presents a novel mechanism of regulation of PD-L1 expression with potentially clinically translatable results.

Reviewer #3 (Remarks to the Author):

This is a potentially interesting study that associates nucleophosmin and PARP1 in the regulation of PD-L1 expression in breast cancer cells. While the data are generally consistent with the conclusions drawn, my major concern is that given the huge abundance of NPM1 protein in cancer cells along with the fact that the protein has been implicated in a huge number cellular processes over the years, one needs to interpret experimental results with the protein cautiously. The authors should provide thorough, complementary data to ensure that the interpretation of the data is correct in order to provide rigor and ensure reproducibility of the key conclusions that are drawn. Thus I have 5 suggestions for additional controls that the authors should consider including in the study to make it fully convincing:

1. Fig. 2D/E: Given the abundance of NPM in tumor cells, its association with anything in pull down assays must be carefully assessed as it is so much more abundant than other nuclear proteins. The miniscule 1.5X fold enrichment of the protein on the biotinylated 469-690 DNA fragment is not very compelling given the abundance of the protein. Furthermore, NPM1 has been implicated as a chaperone for other transcription factors as well as in post-transcriptional control of gene expression. Therefore I would optimally like to see (1) some additional identification of the NPM1-response element in this promoter and (2) an assessment of the loss of NPM1 influence when it is knocked out to firmly establish NPM1 as a bona fide transcription factor in this system.
2. Fig. 2F: Two key controls – non-specific siRNAs and a concurrent assessment of unrelated promoters – are missing and should be added to these experiments.
3. Fig. 2M: While the authors claim on Pg. 5 that PD-L1 was overexpressed with NPM1 KD, the PD-L1 lanes show no change in the data +/- NPM1 KD. Please address this discrepancy between the conclusions and the data.
4. Fig. 5C: Was the NPM1-PARP interaction nucleic acid dependent? RNA bridging should be ruled out in the experiment by treating the samples with RNase prior to antibody pulldown.

Minor points

1. Line 155-157: Results should be reported in the past tense (...decrease (decreased) the tumor size...; ..and reduce (reduced) the occurrence rate...;)
2. Line 209: add a space between figure and 5F

Reviewer #4 (Remarks to the Author):

Summary: Authors evaluate the role of NPM1 in PD-L1 expression in TNBC and whether this interaction has potential as a therapeutic target. They assess PD-L1 expression in clinical TNBC vs. non-TNBC breast cancer cases, and claim that PD-L1 positivity is associated poorer prognosis specifically for TNBC patients. Authors also interrogate if NPM1 promotes PD-L1 expression, and that its knockdown increases T cell activity in vivo. They also evaluate the role of PARP1 in this process, and suggest that PARP1 interacts with NPM1 and limits NPM1-mediated transcription of PD-L1. Finally, authors evaluate whether PARP1 inhibition can improve the efficacy of anti-PD-L1 therapy in a murine model of TNBC by increasing PD-L1 expression by tumor cells. While the idea of NPM1/PDL1 interaction and their role in T cell activity in TNBC is interesting, there are several problem with the study design, interpretation of the data and inconsistent results that have deterred enthusiasm for this study.

Major comments:

- Problems with statistical analyses throughout. Statistical tests are not precisely described, and it is often unclear what comparisons have been made (e.g., does lack of a significance indication imply that the comparison was not performed or that the comparison is insignificant?). Multiple student t-tests are performed in several instances.
- Lines 94-95: "... [PD-L1 positive rate] was in significant inverse correlation with hormone receptor status"
 - o Unclear how this statement is supported by supplementary table 1. A description of what statistical tests / correlations were performed would be helpful. As shown, it is unclear what the statistical designations are reporting (comparison between TNBC and non-TNBC PD-L1 rates?).
- Figure 2H&I: Marginal differences in PD-L1 expression upon NPM1 or (HCC1806) are considered meaningful in (H) but small differences in PD-L1 expression upon IFN-gamma addition in (I) are disregarded.
- Figure 3C: Student's t-test is not appropriate when performing multiple comparisons. ANOVA + correction for multiple comparisons would be more appropriate. More precise description of statistical comparisons is needed. As drawn, it is difficult to determine what comparisons the significance symbols correspond to. Indication of standard deviation on compiled graph would be helpful.
- Figure 3D: Is the difference between (sh-NPM1-nc + PD-L1 vec) and (sh-NPM1+PD-L1 vec) statistically and experimentally meaningful? Was a blinded reader assessing lung metastatic nodules?
- Figure 4: Not clear how the IHC for NPM1 and PDL1 expression were performed (+ve and -ve controls are missing), how the cutoff was determined for each? What is the progression free survival, disease free survival, overall survival and metastatic rate determined for each biomarker? Very little descriptions of the patient cohort was provided including if these were treatment naive samples? What type of treatments did each patient receive.
- Figure 5G: what are the standards for claiming protein levels have increased? Differences presented are not meaningful.
- In vivo experiments combining PARP1 inhibition with anti-PD-L1 fail to demonstrate substantial (meaningful) benefit vs. anti-PD-L1 alone (Figure 7). Discussion of risks associated with PARP1 inhibition (e.g, risk of un-salvageable immunosuppression due to elevated PD-L1 expression) are not discussed. For example, Figure 3D shows that PD-L1 over expression alone causes trend towards increase number of lung metastatic nodules (and rate of metastatic development), which is a concern regarding treatment with PARP1 inhibition that results in increased PD-L1 expression. This would correspond with the worse prognosis of patients who display high PD-L1 as discussed in the introduction (Lines 39-42).
 - o Results in Figure 3C (knockdown of NPM1 suppresses tumor growth) seems to contradict therapeutic approach taken in Figure 7 (inhibition of NPM1's inhibitor).
- Figure 7E: the difference in number of metastases and metastatic rate between anti-PD-L1 monotherapy and the combination therapy appears marginal. Please support your claim that metastasis is best suppressed in combination group with statistical test. Metastasis data actually appears quite similar between all groups.
- Line 267 and Figure 7D: Authors claim that combination therapy was synergistic. Was synergism explicitly tested? If not, the effect could simply be additive. The Spider plots show major heterogeneity of response in each treatment arm and the statistical analysis do not account for this differences in response.

- Figure 7F: Not clear if combination therapy increases functional T cell presence compared to anti-PD-L1 alone. Further, does olaparib alone decrease T cell presence compared to control (this would help determine if there is a substantial added risk of creating a more immunosuppressive environment with olaparib treatment).

- Little mention of Figure 7 in vivo results and their significance

- Lines 382-384: It is discussed that NPM1 is a potential target for TNBC treatment as NPM1 expression is correlated with PD-L1 expression and poor prognosis. This implies a therapeutic strategy to inhibit the PD-L1-promoting activity of NPM1 to overcome TNBC immune evasion. However, this is precisely opposite from the therapeutic strategy undertaken in vivo, where PARP1 was inhibited, resulting in increased PD-L1 expression and modestly improved efficacy of anti-PD-L1 therapy. This is a nuanced problem, and further discussion of the rationale and risks associated with inhibiting PARP1 (increasing NPM1 activity) would be most helpful for the reader. (e.g., is there a risk of PARP1 inhibition causing immunosuppression that is unsalvageable by anti-PD-L1 therapy in some animals? What can be done to mitigate this risk? Would it be more therapeutically beneficial to inhibit NPM1 to overcome one pathway of immunosuppression in TNBC?)

Materials and methods

- Confocal immunofluorescence assay: what controls were used for staining

- Line 546 (Animals and treatment):

- o How many dimensions were measured and what assumptions were made for volume calculation (what equation was used?)?

- o Provide justification for drug dosing. How does olaparib dose compare to what would be used clinically?

- Line 595 (Histopathology): Were pathologists blinded?

- Line 599 (Data analysis): Statistical tests should be described in more detail. ANOVA with appropriate multiple comparisons testing should be performed instead of student's t-test wherever >2 groups are being compared. Justification/description of how correlation was performed would be helpful (what is Pearson χ^2 -test, and is this appropriate for drawing correlations with categorical data? Would a chi-squared test be more appropriate?)

Manuscript ID: NCOMMS-18-33300A

Title: “NPM1 upregulates the transcription of PD-L1 and suppresses T-cell activity in triple negative breast cancer”

Responses to Reviewers

To Reviewer #1:

1. The figures have too many panels that can be placed in the supplement and only a summary panel provided in the main figure.

Response: We have rearranged the figures, and some panels in figure 2, figure 3, figure 5 and figure 7 have been moved to supplementary figures.

2. In supplementary table 1, it is unclear what the significant P values are referring to. For example, when considering tumor size, what is driving the significant p-Value? Is it the fact that a tumor is less likely to be PDL1 negative if it is <20 mm? This needs to be defined for all the characteristics listed where there is significance.

Response: Thanks to the reviewer for the valuable suggestion. “PD-L1 positive rate in TNBC was 61.5% (32/52), but was only 18.6% (18/97) in non-TNBC (figure 1B, supplementary table 1). In addition, tumors in larger volume (diameter >20 mm) had a higher positive rate, which was in significant inverse correlation with hormone receptor (HR) status (supplementary table 1). This description has been added to “Result - TNBCs have higher PD-L1 expression” part. The correlational relationship was also marked in supplementary table 1. (line 93-97)

3. In Supplementary table 1, what is the median age for the PDL1 positive group?

Response: The median age for the PD-L1 positive group is 47.5.

4. Can the Kaplan Meir survival curves be broken up by stage and PDL1 status?

Response: We added the Kaplan Meir survival curves for PD-L1 in early stage (phase I) and middle stage (phase II~III) breast cancer patients. The result showed that PD-L1 positive patients had significantly poorer prognosis in the early stage cohort. However, PD-L1 expression did not significantly influence the OS in middle stage patients.

5. In Figure 2 D, were other proteins identified within the “NPM1” band that was cut out for mass spectrometry?

Response: There was no other protein identified in this band, but two other proteins around 40KD were identified (see below).

1	P0CG38	198	POTE ankyrin domain family member I OS=Hom o sapiens GN=POTEI PE=3 SV=1
2	P60709	198	Actin, cytoplasmic 1 OS=Hom o sapiens GN=ACTB PE=1 SV=1

6. Representative mass spectrometry data must be provided in the supplement.

Response: The representative MS data has been included in supplementary figure 1A.

7. Figure 2 has too many panels that should be placed in the supplement.

Some of the data is redundant. It should be included, but not in the primary figure.

Response: The original figure 2I and 2J have been moved to supplementary figure 1C and 1D.

8. What was the effect of sh-NPM1 on the surface expression of PD-L1 in the other cell lines? All this data, including Fig.2J, can be shown in the supplement.

Response: The original figure 2J has been moved to supplementary figure 1D. The data from HCC1806 cells have also been provided in supplementary figure 1D.

9. Why do the authors only focus on a couple of cell lines for further studying NPM1 and PDL1, when they initially introduce a number of cell lines? It is perfectly acceptable to study only 1 or 2 of the cell lines, but a rationale needs to be provided. Or alternatively, if the authors performed the experiments in all the TNBC cell lines, then they should show these data in the supplement.

Response: Thanks to the review's kind suggestion. We initially used MDA-MB-231, HCC-1806, HCC1937, BT20 and HS578T cells to analyze the endogenous expression of PD-L1 in triple negative breast cancer. Among them, HCC1937 is a BRCA1 mutant cell line, BT20 typically over expresses WNT3 and WNT7B oncogenes and Hs578t is a fibroblast-like cell line. These three cell lines may have stronger heterogeneity. Therefore, MDA-MB-231 and HCC1806 cell lines are more representative for TNBC and were used in the further studies. The relevant description has been provided in "Result-NPM1 binds to PD-L1 promoter and upregulates PD-L1 expression in TNBC cells" part. (line 126-130)

10. The use of jurkat cells as the effector cells to show the effects of NPM1/PDL1/PARP on immune cell functions needs to be justified. Ideally, the authors should expand T cells (from healthy donor peripheral blood) that react

with the tumor cell line using a known tumor antigen for the cognate cell line or using tumor cell lysates. Is there data to show that Jurkat cells recognize MDA-MB-231 and HCC1806? If so, references in well-respected journals should be provided to support the use of this model.

Response: Thanks to the reviewer for the valuable suggestion. PD-L1 expression level is associated with the activity of Jurkat cells. Jurkat cells are able to recognize the PD-L1 expression of multiple kinds of tumor cells including lymphoma cells, melanoma cells, lung cancer cells and breast cancer cells (Reference 17, 18, 27, 28) . To further verify our research, B16-OVA cells were co-cultured with OT-I mouse splenic lymphocytes, which recognized OVA antigen specifically (Supplementary figure 1E).

11. In figure 3, the authors need to state what PDL1-vec means. Is this the negative control vector for PDL1?

Response: “4T1 cells were stably transfected with control shRNA (sh-NPM1-nc) or NPM1 shRNA (sh-NPM1) and the control vector for PD-L1 (PD-L1 vec) or PD-L1 overexpression plasmid (PD-L1 oe).” This statement has been added to the legend of figure 3. (line 1012-1014)

12. Figure 3: too many panels. Much of the data can be included in the supplement. For example, Figure 3 C is the summary figure and should be included, while 3B should be in the supplement.

Response: The original figure 3B has been moved to supplementary figure 2E.

13. The authors should discuss briefly cold vs. hot tumors as they relate to PDL1 expression and TNBC.

Response: Thanks for the kind suggestion. “Tumor infiltrating lymphocytes (TILs) have been associated with improved prognosis in many different tumor types, as the paucity of tumor T cell infiltration often leads to initial resistance

to immunotherapy. Tumors that lack TILs were characterized as "cold tumors", whereas tumors with massive T cell infiltration were defined as "hot tumors". Though each subtype of breast cancer has both "cold" and "hot" tumors, TNBCs often have tumors with >50 % lymphocytic infiltration and have higher PD-L1 expression, which indicate that TNBCs may be the most sensitive to anti-PD-L1 therapy among breast cancer." This has been added to the "Discussion" part. (line 314-322)

14. Did the authors look at the expression of cytokines and chemokines in the tumor tissue that may have contributed to the increase in T cell infiltration in the in vivo model? If so, please show data. If not, please discuss why you think the decrease in NPM1 and PDL1 contributed to the increase in TIL.

Response: In our research, we did not explore all kinds of TILs but mainly focused on the CD8+ T cells. The interaction of PD-1 expressed on CD8+ T cells and PD-L1 expressed on tumor cells plays the key role in immune escape. The activation of the PD-1/PD-L1 axis restricts CD8+ T cell expansion and inhibits its anti-tumor activity. We found that NPM1 knockdown decreased CD8+ T cell ratio and activity indicated by CD69, CD107, Granzyme B, and these effects were reversed by PD-L1 overexpression. Based on these evidences, we think decrease in NPM1 and PDL1 contributes to the increase in amount and activity of CD8+ T cells. The relevant description has been included in "Results- Knockdown of NPM1 suppresses PD-L1 expression and promotes T cell activity *in vivo*" section of the revised manuscript. (line 178-181, 187-189)

15. The scientific studies showing that NPM1 regulates PDL1 are convincing. However, the claim that NPM1 is an independent poor clinical prognosticator in TNBC is inappropriate to state without performing multivariate analyses to demonstrate this fact. The authors should CLEARLY state that their results are based on univariate, possibly biased analyses and MUST be confirmed in larger cohorts with appropriate statistical analyses.

Response: We conducted the multivariate survival analysis using COX-regressive analysis (Supplementary table 3). Although NPM1 was not an independent prognostic factor based on our COX-regression analysis, our analysis showed that patients with negative hormone receptor expression tended to have higher level of NPM1 (supplementary table 4), suggesting that NPM1 still had prognostic value in breast cancer treatment. This result will be confirmed in a larger cohort and perhaps different kinds of solid tumors as well in our future research. The statement has been added to “Result- NPM1 is a poor prognostic factor and positively correlated with PD-L1 expression in breast cancer” part. Thanks for the valuable comments. (line 204-209)

16. The patient characteristics of the cohort that was used in Fig. 4 need to be published in the supplemental material.

Response: The patient characteristics of the cohort were provided in supplementary table 4.

17. Figures 4C and D: Which comparisons are driving the statistical significance?

Response: Pearson Chi-Square was used to do the correlation analysis. (line 1032-1033)

18. There is no information in the methods on mass spectrometry. This needs to be added.

Response: The MS information has been added in methods.

19. Figure 5A: there is no band adjacent the arrow indicating PARP.

Response: We are sorry for this mistake. The band of PARP is about 100KD. We have corrected the arrow position.

20. Figure 5 G Figure legend states NPM1 knockdown or overexpression.

Should this be PARP1 (not NPM1)?

Response: We are sorry for this carelessness. The mistake has been corrected.

21. A reference needs to be added showing that olaparib inhibits MOUSE PARP1.

Response: We used a PARP1 activity measurement kit to detect the variation of PARP1 enzymatic activity in olaparib treated 4T1 cells. The result was shown in supplementary figure 5K.

22. Fig. 7: the authors should demonstrate the effects of olaparib on T cell functions, including T cell homing and activation, and expression of conventional T cell surface markers. This can be done in in vitro assays.

Response: We thank the reviewer for the nice suggestion. We has proved that olaparib had no effect on lymphocytes homing and activation in vitro (supplementary figure 5I, J).

Minor comment:

1. The word “Besides” should be replaced with a more appropriate synonym throughout the manuscript.

Response: We have changed “besides” to “furthermore”, “moreover”, “also”, etc. in the manuscript.

2. Need to define “oe” and “nc” in the figure 2 legend.

Response: “PD-L1 promoter activity in MDA-MB-231 cells transiently transfected with control siRNA (nc), NPM1 siRNAs (si-NPM1-1, si-NPM1-2), control plasmid for NPM1(vec), or NPM1 overexpression plasmid (NPM1 oe) was measured by dual-luciferase assay.” This statement was included in the legend of figure 2F. (line 986-989)

3. *Some of the labels for many figures, (example Figure 5G) are cut off.*

Response: These figures have been corrected.

4. *Figure 5E in the HC1806 data, PARP is spelled as PAPR.*

Response: This misspelling has been corrected.

5. *Figure 5G PARP is mis-spelled.*

Response: This mistake has been corrected.

6. *Figure 6: some of the labels are cut off.*

Response: These figures have been corrected.

To Reviewer #2:

1. *NPM1 is an abundant protein, not tissue specific, and is known to interact with many different proteins. Can the authors show a direct interaction of NPM1 and PARP-1 in a cell-free system?*

Response: Thanks to the reviewer for the valuable suggestion. We have purified Flag-NPM1 and Myc-PARP1 proteins and proved their interaction in a cell-free system. The results were shown in figure 5D and supplementary figure 5C.

2. *The authors show that PARP-1 interacts with NPM1 and suppresses the expression of PD-L1. Have they tested if PARP inhibitors disrupt the interaction of NPM1 and PARP-1?*

Response: We have confirmed that PARP inhibitor olaparib disrupts the interaction of NPM1 and PARP1. The result is shown in supplementary figure

5G.

3. *Do NPM1 inhibitors, such as NSC348884, reduce TNBC xenograft growth and increase the infiltration/activity of CD8+ T cells?*

Response: The NPM1 inhibitor NSC348884 has a specific effect on NPM1. NSC348884 prevents the formation of NPM1 oligomer but has no effect on NPM1 expression. We noticed that NSC348884 did not decrease NPM1 protein level or PD-L1 expression in MDA-MB-231 and 4T1 cells (supplementary figure 7A). NSC348884 significantly inhibited tumor growth *in vivo* (supplementary figure 7B, C). However, it neither decreased PD-L1 expression nor increased CD8+ cell infiltration *in vivo* (supplementary figure 7D, F). These results indicated that the anti-tumor effect of NSC348884 depended on its anti-proliferation effect, and NPM1 regulated PD-L1 transcription in its monomer form. These comments have been added in the "Discussion" part. (line 381-390)

4. *The authors show that NPM1 expression is positively correlated with PD-L1 expression in approximately 57% of TNBC samples. Could they comment on the possible reasons for high PD-L1 expression in samples that do not have high expression of NPM1?*

Response: There are two reasons that may contribute to NPM1 low/PD-L1 high samples. First, PD-L1 expression is not only associated with the total expression of NPM1, but also its active form. As we had mentioned, NPM1 can undergo extensive post-translational modifications including phosphorylation, acetylation, ubiquitination, and SUMOylation, which are likely to control its stability, localization, interaction with other proteins and the related cellular functions. While NPM1 mutant is rarely detected in solid tumors, we speculate that a certain NPM1 modification plays a crucial role in upregulating PD-L1 transcription. This will be further explored in our future work. Second, TNBCs have the strongest heterogeneity. Though ER-/PR-/HER-2- breast cancers are all defined as TNBCs, there are different biological behaviors, clinical features, therapeutic responses and driver genes among them, and the function of NPM1 in regulating PD-L1 expression may

be related to the subtypes of TNBCs. We have discussed this issue in the Discussion section of the revised manuscript. (line 391-405)

To Reviewer #3 :

1. Fig. 2D/E: Given the abundance of NPM in tumor cells, its association with anything in pull down assays must be carefully assessed as it is so much more abundant than other nuclear proteins. The miniscule 1.5X fold enrichment of the protein on the biotinylated 469-690 DNA fragment is not very compelling given the abundance of the protein. Furthermore, NPM1 has been implicated as a chaperone for other transcription factors as well as in post-transcriptional control of gene expression. Therefore I would optimally like to see (1) some additional identification of the NPM1-response element in this promoter and (2) an assessment of the loss of NPM1 influence when it is knocked out to firmly establish NPM1 as a bona fide transcription factor in this system.

Response: Thank you for pointing out the problem in our study. It was reported that NPM1 recognized and directly bound to a G-quadruplex sequence at the c-Myc promoter. However, we did not find a specific G-quadruplex sequence in the PD-L1 promoter region. In our research, we demonstrated that NPM1 was a crucial transcription regulator of PD-L1 and it might exert its activity by interacting with other proteins, undergoing epigenetic modifications or acting as a transcription factor (as we have mentioned in “Discussion”). However, to rigorously prove that NPM1 is a bona fide transcription factor exceeds the scope of the current study, and we will address this in the follow-up study. (line 374-377)

2. Fig. 2F: Two key controls – non-specific siRNAs and a concurrent

assessment of unrelated promoters – are missing and should be added to these experiments.

Response: Thanks for the suggestion. We used a non-specific siRNA as negative control (nc) in these experiments. In addition, we have used an APCL siRNA and detected PD-L1 promoter activity. APCL is an APC family gene that mainly expresses in brain tissue but barely expresses in mammary tissue. Furthermore, we used a FOP promoter, which contains a mutant β -catenin targeted promoter region, as an irrelevant promoter. These results were presented in supplementary figure 1B.

3. Fig. 2M: While the authors claim on Pg. 5 that PD-L1 was overexpressed with NPM1 KD, the PD-L1 lanes show no change in the data +/- NPM1 KD. Please address this discrepancy between the conclusions and the data.

Response: When endogenous NPM1 was knocked down, the endogenous PD-L1 expression was decreased. On the other hand, when PD-L1 was exogenously overexpressed, knockdown of endogenous NPM1 was insufficient to completely abolish the expression of PD-L1.

4. Fig. 5C: Was the NPM1-PARP interaction nucleic acid dependent? RNA bridging should be ruled out in the experiment by treating the samples with RNase prior to antibody pulldown.

Response: We have purified Flag-NPM1 and Myc-PARP1 proteins and proved their interaction in a cell-free system. The results are shown in figure 5D and supplementary figure 5C.

Minor points

1. Line 155-157: Results should be reported in the past tense (...decrease (decreased) the tumor size....; ..and reduce (reduced) the occurrence rate...;)

Response: The tense has been corrected.

2. Line 209: add a space between figure and 5F

Response: The mistake has been corrected.

To Reviewer #4:

Major comments:

1. *Problems with statistical analyses throughout. Statistical tests are not precisely described, and it is often unclear what comparisons have been made (e.g., does lack of a significance indication imply that the comparison was not performed or that the comparison is insignificant?). Multiple student t-tests are performed in several instances.*

Response: Thank you for pointing out the problem in our study. We have reanalyzed the multiple group data using ANOVA analysis. The statistical methods have been described in each figure legend and Methods-Statistics part.

2. *Lines 94-95: "... [PD-L1 positive rate] was in significant inverse correlation with hormone receptor status" Unclear how this statement is supported by supplementary table 1. A description of what statistical tests / correlations were performed would be helpful. As shown, it is unclear what the statistical designations are reporting (comparison btwn TNBC and non-TNBC PD-L1 rates?).*

Response: "Pearson chi-square analysis was used to determine the correlation between PD-L1 expression and other clinical features." This statement has been added to "Result - TNBCs have higher PD-L1 expression" part (line 92-93). Besides, we have labeled the positive or negative correlation in supplementary table 1. Thanks for the valuable suggestion.

3. *Figure 2H&I: Marginal differences in PD-L1 expression upon NPM1 oe (HCC1806) are considered meaningful in (H) but small differences in PD-L1*

expression upon IFN-gamma addition in (I) are disregarded.

Response: IFN- γ is an important PD-L1 inducing factor It is acceptable that PD-L1 expression was upregulated under IFN- γ induction. Therefore, we have improved the description to “As shown in supplementary figure 1C, knockdown of NPM1 also decreased PD-L1 expression in the presence of IFN- γ , suggesting that NPM1 was a dominant regulatory factor for PD-L1 expression in TNBC cells.” (line 142-143)

4. Figure 3C: Students t-test is not appropriate when performing multiple comparisons. ANOVA + correction for multiple comparisons would be more appropriate. More precise description of statistical comparisons is needed. As drawn, it is difficult to determine what comparisons the significance symbols correspond to. Indication of standard deviation on compiled graph would be helpful.

Response: Thank you for the advice. We have reanalyzed all the multiple comparisons in this study using ANOVA + correction analysis. The significance symbols have been presented in a clearer way in figure 3C.

5. Figure 3D: Is the difference between (sh-NPM1-nc + PD-L1 vec) and (sh-NPM1+PD-L1 vec) statistically and experimentally meaningful? Was a blinded reader assessing lung metastatic nodules?

Response: When PD-L1 expression was strongly overexpressed by lentivirus, endogenous NPM1 knockdown had slight effect on PD-L1 expression. The pathologists were blinded.

6. Figure 4: Not clear how the IHC for NPM1 and PDL1 expression were performed (+ve and -ve controls are missing), how the cutoff was determined for each? What is the progression free survival, disease free survival, overall survival and metastatic rate determined for each biomarker? Very little descriptions of the patient cohort was provided including if these were treatment naïve samples? What type of treatments did each patient receive.

Response: Two slides of tumor tissue microarray were used to analyze

NPM1 and PD-L1 expression in figure 4. We have analyzed the correlation between NPM1 expression and distant metastasis in supplementary table 1. However, PFS and DFS data were not provided with the microarray. We have added the description “These tissues were all obtained at their first operation with no previous treatment. The therapeutic regimen of these patients was not provided” to “Materials and Methods - Human tissue specimens” part. (726-728)

We have also added the description of NPM1 and PD-L1 scoring to “Materials and Methods - Histopathology” part. “PD-L1 positive was defined as positive cells >1%. NPM1 expression was scored by the staining intensity and percentage of positive cells. The intensity was classified into four scores: “0” for no brown particle staining, “1” for light brown particles, “2” for moderate brown particles, and “3” for dark brown particles. The percentage of positive cells was also divided into four scores: “0” for <10% positive cells, “1” for 10%–40% positive cells, “2” for 40%–70% positive cells, and “3” for ≥70% positive cells. The two scores were multiplied and used to determine high (score ≥ 3) or low (score < 3) expression of NPM1.” (line 736-744)

7. Figure 5G: what are the standards for claiming protein levels have increased? Differences presented are not meaningful.

Response: We have replaced the figure with one of our repeated experiment. The grayscale was measured by Image J.

7. In vivo experiments combining PARP1 inhibition with anti-PD-L1 fail to demonstrate substantial (meaningful) benefit vs. anti-PD-L1 alone (Figure 7). Discussion of risks associated with PARP1 inhibition (e.g, risk of un-salvageable immunosuppression due to elevated PD-L1 expression) are not discussed. For example, Figure 3D shows that PD-L1 over expression alone causes trend towards increase number of lung metastatic nodules (and rate of metastatic development), which is a concern regarding treatment with PARP1 inhibition that results in increased PD-L1 expression. This would correspond with the worse prognosis of patients who display high PD-L1 as discussed in

the introduction (Lines 39-42).

Response: According to the recent reports, the immunoregulation activity of olaparib has its two sides. the immunoregulation activity of olaparib is still controversial. On the one hand, as is shown by us here and others, olaparib enhances PD-L1 expression, while on the other hand, olaparib has also been reported to stimulate CD8+ T cell infiltration by upregulating the STING pathway. So far, there has been no evidence that olaparib will promote tumor growth, increase metastatic rate, or induce irreversible immunosuppression by upregulating PD-L1 expression. It is conceivable that the dominant effect of olaparib is its cytotoxicity, and the immunosuppressive activity caused by increased PD-L1 expression is probably a critical factor that weakens the anti-tumor effects of olaparib. The relevant demonstration has been added to the “Discussion” part. (line 444-453)

8. Results in Figure 3C (knockdown of NPM1 suppresses tumor growth) seems to contradict therapeutic approach taken in Figure 7 (inhibition of NPM1's inhibitor).

Response: Olaparib, the compound used in figure 7, is a PARP inhibitor but not a NPM1 inhibitor. This result was not in conflict with Figure 3C and it was consistent with Figure 5.

9. Figure 7E: the difference in number of metastases and metastatic rate between anti-PD-L1 monotherapy and the combination therapy appears marginal. Please support your claim that metastasis is best suppressed in combination group with statistical test. Metastasis data actually appears quite similar between all groups.

Response: The description has been modified to “We also observed that the pulmonary metastasis rate and the number of lung metastatic nodules were the lowest in the combination therapy group, though not significantly lower than the monotherapy groups.” and this figure has been moved to supplementary figure 6D. (line 295-298)

10. Line 267 and Figure 7D: Authors claim that combination therapy was synergistic. Was synergism explicitly tested? If not, the effect could simply be additive. The Spider plots show major heterogeneity of response in each treatment arm and the statistical analysis do not account for these differences in response.

Response: We are sorry for our inaccurate description. Ideally, the synergism of combination therapy *in vivo* should have been tested with different dosages and analyzed with a CI curve. However, since the combination regimen had been reported before (as commented in the “Discussion” part), we did not explicitly test the synergism in our study. Accordingly, we have corrected the description. (line 309-311)

11. Figure 7F: Not clear if combination therapy increases functional T cell presence compared to anti-PD-L1 alone. Further, does olaparib alone decrease T cell presence compared to control (this would help determine if there is a substantial added risk of creating a more immunosuppressive environment with olaparib treatment).

Response: There was no significant differences between combination therapy and anti-PD-L1 monotherapy on CD8+ T cell infiltration. In our view, the anti-tumor effect of olaparib mainly depends on its cytotoxic effect. However, olaparib induced PD-L1 expression may cause immunosuppression and reduce the anti-tumor effect of olaparib. Consistently, we observed the tendency that olaparib alone had lower CD8+T cell infiltration and activity than control, though it was not statistically significant. As we have mentioned before, the immunoregulation mechanism and effect of olaparib are still controversial and need to be further explored. Nonetheless, the combination of olaparib and immunotherapy is promising and reasonable.

12. Little mention of Figure 7 *in vivo* results and their significance

Response: Olaparib monotherapy has proved superior to single-agent chemotherapy with an increase of mPFS from 4.2 to 7.0 month but no significant improvement in overall survival. Studies have also indicated that

combination therapy of olaparib and platinum chemotherapy could be more beneficial. However, chemotherapies are frequently accompanied with side effects that can lead to incompliance. In our study, we found olaparib increased PD-L1 expression by interacting with NPM1 *in vivo* and *in vitro*. The immunosuppressive action of PD-L1 might be an important factor that weakens olaparib monotherapy effects. Our *in vivo* study showed that olaparib and anti-PD-L1 combination therapy had better therapeutic effect than monotherapy in BRCA1 deficient TNBC xenografts. This result may provide a novel combination regimen for TNBC with lower toxicity than the olaparib and chemotherapy combination. The relevant description has been included in "Discussion" part. (line 430-436, 453-458)

*13. Lines 382-384: It is discussed that NPM1 is a potential target for TNBC treatment as NPM1 expression is correlated with PD-L1 expression and poor prognosis. This implies a therapeutic strategy to inhibit the PD-L1-promoting activity of NPM1 to overcome TNBC immune evasion. However, this is precisely opposite from the therapeutic strategy undertaken *in vivo*, where PARP1 was inhibited, resulting in increased PD-L1 expression and modestly improve efficacy of anti-PD-L1 therapy. This is a nuanced problem, and further discussion of the rationale and risks associated with inhibiting PARP1 (increasing NPM1 activity) would be most helpful for the reader. (e.g., is there a risk of PARP1 inhibition causing immunosuppression that is unsalvageable by anti-PD-L1 therapy in some animals? What can be done to mitigate this risk? Would it be more therapeutically beneficial to inhibit NPM1 to overcome one pathway of immunosuppression in TNBC?)*

Response: We evaluated the effect of a NPM1 inhibitor, NSC348884.

NSC348884 has a specific effect on NPM1, as it prevents the formation of NPM1 oligomer but has no effect on NPM1 expression. We noticed that NSC348884 did not decrease NPM1 protein level or PD-L1 expression in MDA-MB-231 and 4T1 cells (supplementary figure 7A). NSC348884 significantly inhibited tumor growth *in vivo* (supplementary figure 7B, C). However, it neither upregulated CD8⁺T cell infiltration nor decreased PD-L1

expression *in vivo* (supplementary figure 7D, F). These results indicated that the anti-tumor effect of NSC348884 depended on its anti-proliferation effect, for NPM1 oligomer plays an important role in cell proliferation, and NPM1 regulated PD-L1 transcription in its monomer form. As we have mentioned before, the immunoregulation activity of olaparib is complicated. There is no evidence that olaparib can induce irreversible immunosuppression. The relevant comments have been added to the “Discussion” part. (line 381-390)

Materials and methods

14. *Confocal immunofluorescence assay: what controls were used for staining*

Response: %1 BSA in PBS was included as a negative control when cells were incubated with primary antibodies. (line 623-625)

15. *Line 546 (Animals and treatment): How many dimensions were measured and what assumptions were made for volume calculation (what equation was used?)?*

Response: Two dimensions were measured and tumor volume (TV) was calculated as $TV \text{ (mm}^3\text{)} = \pi/6 \times \text{length} \times \text{width}^2$. (line 687-688)

16. *Provide justification for drug dosing. How does olaparib dose compare to what would be used clinically?*

Response: The clinically used dosage of olaparib is 300 mg twice a day. Assuming that the average weight of patients is 60 kg, the dosage is 10 mg/kg a day. We used 50 mg/kg olaparib in our research, which is a common dosage in mouse models.

17. *Line 595 (Histopathology): Were pathologists blinded?*

Response: The pathologists were blinded. The relevant description has been added. (line 736)

18. Line 599 (Data analysis): Statistical tests should be described in more detail. ANOVA with appropriate multiple comparisons testing should be performed instead of students t-test wherever >2 groups are being compared. Justification/description of how correlation was performed would be helpful (what is Pearson w2-test, and is this appropriate for drawing correlations with categorical data? Would a chi-squared test be more appropriate?)

Response: The “Data analysis” part has been improved with more accurate description of statistical methods. (line 748-752)

Reviewers' comments:

Reviewer #1 (Remarks to the Author):

Thank you for addressing my concerns.
I have two minor concerns:

1. Please include the median age (47.5) for the PD-L1 positive group in the table. You answered my inquiry but did not put it in the table.
2. Please include the Kaplan Meir curves by stage and PD-L1 status to the manuscript supplement. You responded to my inquiry but did not include them in the manuscript.

Thank you.

Reviewer #3 (Remarks to the Author):

While I find the revised manuscript to be improved, I was dismayed to see that the authors failed to address two of the four major points (# 1 and 4) from my original critique. Addressing both of these points would have substantially increased the rigor of the study and provide stronger support for the major conclusions of the manuscript.

Other Points:

1. As stated above, Fig. 5D still needs RNase treatment to establish a direct interaction. Nucleic acids will routinely purify with RNA binding proteins.
2. Fig. 2F: I'm curious as why PDL-1 promoter activity varies so much figure to figure in this study. It functions in control experiments in the 300 range in 2F (left), 50 range (Fig 2F right) and even lower values in the control samples in later figures in the manuscript. This variation highlights in my opinion the need for the rigorous validation requested in point 1 of my original critique.

Reviewer #4 (Remarks to the Author):

The finding that NPM1 regulates PD-L1 and associated immunosuppression in TNBC is novel and important. However, the fact that NPM1 inhibition does not result in therapeutic benefit through the immunologic mechanism described in through the rest of the paper decreases enthusiasm about the manuscript and its translational significance. Thus, I do not recommend this manuscript for acceptance into Nature Communications.

Combination of PARPi and anti-PD-L1 does show modest benefit through expected immunological mechanism. However, this combination is not novel as it has been proposed and tested previously in preclinical setting (– citation 58 – as discussed in Discussion section).

Remarks regarding the adequacy of individual rebuttals and/or remaining concerns are shown below as comments linked to each individual point that has not yet been addressed.

- 1) Figure 3C: Student's t-test is not appropriate when performing multiple comparisons. ANOVA + correction for multiple comparisons would be more appropriate. More precise description of statistical comparisons is needed. As drawn, it is difficult to determine what comparisons the significance symbols correspond to. Indication of standard deviation on compiled graph would be helpful. Also text lines 187-189: I think they may mean to say "that increased activity of CD8+ T cells is reversed by PDL1 O.E.? Unfortunately, the direct comparison of (shNPM1 + PDL1 vector) vs

(shNPM1 + PD-L1 OE) does not appear to be significant.

2) Figure 3D-Not answering about lung metastatic nodules (original Figure 3D)

3) In vivo experiments: Still a concern since the purpose of this figure is to show that anti-PD-L1 can improve PARPi since increased PDL1 expression is an unfortunate consequence of PARPi. Combination therapy effect is still not very impressive, especially when you see how large the variability is between groups (in supplemental figure).

Additional concern: why is the treatment schedule for aPD-L1 alone different than when it is combined with PARPi (see schedules in Figure 7A).

4) The purpose of Figure 7 is to show that aPD-L1 can help to overcome the unfortunate increase in PD-L1 expression that you get as a side effect of PARPi. This is independent of the idea that you could target NPM1 therapeutically (with the goal of decreasing immunosuppression).

5) Figure 7D: The combination therapy is still described as synergistic throughout the rest of the manuscript.

The fact that PARPi + aPD-L1 has already been tested as combination therapy decreases the novelty and importance of this figure.

Lines 382-384 of the previous version: The fact that NPM1 inhibition does not result in therapeutic benefit through the immunologic mechanism described in through the rest of the paper decreases enthusiasm about the manuscript and its translational significance.

Combination of PARPi and anti-PD-L1 does show modest benefit through expected immunological mechanism. However, this combination is not novel as it has been proposed and tested previously.

Reviewer #5 (Replacement Reviewer for Reviewer #2, Remarks to the Author):

I would say that the authors have addressed each of the substantive points raised by reviewer 2. The interactions between NPM1 and PARP1 has been shown and this appears to be abrogated by PARP inhibitor. The argument about why the NPM1 inhibitor expt would not necessarily be informative is salient as is the addition of the discussion around PDL1 expression.

Manuscript ID: NCOMMS-18-33300B

Title: “*NPM1 upregulates the transcription of PD-L1 and suppresses T-cell activity in triple negative breast cancer*”

Responses to Reviewers

To Reviewer #1:

1. Please include the median age (47.5) for the PD-L1 positive group in the table. You answered my inquiry but did not put it in the table.

Response: We thank the reviewer for the suggestion. The median age was included in Supplementary Table 1.

2. Please include the Kaplan Meier curves by stage and PD-L1 status to the manuscript supplement. You responded to my inquiry but did not include them in the manuscript.

Response: The Kaplan Meier curves by stage and PD-L1 status were included in Supplementary Figure 1A (Line 103-107) .

To Reviewer #3:

1. As stated above, Fig. 5D still needs RNase treatment to establish a direct interaction. Nucleic acids will routinely purify with RNA binding proteins.

Response: We thank the reviewer for the valuable suggestion. We have proved the direct interaction between the purified Flag-NPM1 and Myc-PARP1 with RNase treatment, and the results were shown in Figure 5D (Line 232-234).

2. Fig. 2F: I'm curious as why PDL-1 promoter activity varies so much figure to figure in this study. It functions in control experiments in the 300 range in 2F (left), 50 range (Fig 2F right) and even lower values in the control samples in later figures in the manuscript. This variation highlights in my opinion the need for the rigorous validation requested in point 1 of my original critique.

Response: Thanks to the reviewer for pointing out the problem in our study. In Figure 2F, the variation of this ratio was most likely due to the transfection protocol. In Figure 2F (left), siRNA was used to transfect cells first. After 24 hours, dual-luciferase reporter plasmids were transfected into cells. The amount of PD-L1 reporter plasmid was 1 μ g per well. However, in Figure 2F (left), NPM1 overexpressing plasmid and dual-luciferase reporter plasmids were transfected at the same time. The ratio of NPM1 overexpressing plasmid and PD-L1 reporter plasmid was 1:1 and their total amount was 1 μ g per well. We have normalized the dual-luciferase assay value in this study. Besides, we also proved a potential DNA binding site of NPM1. It has been reported that NPM1 binds to a G-rich region in DNA with the repetitive sequence "TTAGGG". This sequence was also found in -641 ~ -663 of PD-L1 promoter region. Our result showed that NPM1 overexpression failed to promote PD-L1 transcription activity of -641 ~ -663 deleted PD-L1 promoter, which indicated this region may play an important role (Supplementary Figure 5L). The relevant demonstration was added to the "Discussion" part in the revised manuscript (Line 382-387).

To Reviewer #4:

1. Figure 3C: Student's *t*-test is not appropriate when performing multiple comparisons. ANOVA + correction for multiple comparisons would be more appropriate. More precise description of statistical comparisons is needed. As drawn, it is difficult to determine what comparisons the significance symbols correspond to. Indication of standard deviation on compiled graph would be helpful. Also text lines 187-189: I think they may mean to say "that increased activity of CD8+ T cells is reversed by PDL1 O.E.? Unfortunately, the direct comparison of (shNPM1 + PDL1 vector) vs (shNPM1 + PD-L1 OE) does not

appear to be significant.

Response: We thank the reviewer for the kind advice. Figure 3C was analyzed by one-way ANOVA + Tukey test. The demonstration was corrected to “In addition, the increased ratio and activity of CD8+ T cells caused by NPM1 knockdown were reversed by PD-L1 overexpression” (Line 192-194).

2. Figure 3D-Not answering about lung metastatic nodules (original Figure 3D).

Response: Lung metastatic nodules were analyzed and shown in Figure 3C.

3. In vivo experiments: Still a concern since the purpose of this figure is to show that anti-PD-L1 can improve PARPi since increased PDL1 expression is an unfortunate consequence of PARPi. Combination therapy effect is still not very impressive, especially when you see how large the variability is between groups (in supplemental figure).

Additional concern: why is the treatment schedule for aPD-L1 alone different than when it is combined with PARPi (see schedules in Figure 7A).

Response: We thank the reviewer for the valuable comments. The treatment schedule was referred to other relevant studies. In combination therapy, PARPi was used several days early to promote the expression of PD-L1, which might make tumor more sensitive to anti-PD-L1.

4. The purpose of Figure 7 is to show that aPD-L1 can help to overcome the unfortunate increase in PD-L1 expression that you get as a side effect of PARPi. This is independent of the idea that you could target NPM1 therapeutically (with the goal of decreasing immunosuppression).

Response: According to our research, there is evidence that NPM1 is a potential therapeutic target of TNBC. We have tried to explore the effect of NPM inhibitor NSC348884. However, NSC348884 did not downregulate PD-L1 expression, as NSC348884 prevents the formation of NPM1 oligomer but has no effect on NPM1 expression. It is known that NPM1 can undergo

extensive post-translational modifications, we propose that NPM1 monomer may be uniquely modified in TNBC cells to regulate PD-L1 expression. We are further exploring the active form of NPM1 and find a more effective NPM1 inhibitor in our ongoing work.

5. Figure 7D: The combination therapy is still described as synergistic throughout the rest of the manuscript.

Response: We have corrected all these descriptions in the revised manuscript (Line 18, 83-86, 299-300, 1141-1142).

REVIEWERS' COMMENTS:

Reviewer #3 (Remarks to the Author):

The authors have adequately addressed my lingering concerns. I find the revised manuscript to be improved and convincing.